# Learning on LoRAs:
# GL-Equivariant Processing of Low-Rank Weight Spaces for Large Finetuned Models

## Abstract

Low-rank adaptations (LoRAs) have revolutionized the finetuning of large foundation models, enabling efficient adaptation even with limited computational resources. The resulting proliferation of LoRAs presents exciting opportunities for applying machine learning techniques that take these low-rank weights themselves as inputs. In this paper, we investigate the potential of Learning on LoRAs (LoL), a paradigm where LoRA weights serve as input to machine learning models. For instance, an LoL model that takes in LoRA weights as inputs could predict the performance of the finetuned model on downstream tasks, detect potentially harmful finetunes, or even generate novel model edits without traditional training methods. We first identify the inherent parameter symmetries of low rank decompositions of weights, which differ significantly from the parameter symmetries of standard neural networks. To efficiently process LoRA weights, we develop several symmetry-aware invariant or equivariant LoL models, using tools such as canonicalization, invariant featurization, and equivariant layers. We finetune thousands of text-to-image diffusion models and language models to collect datasets of LoRAs. In numerical experiments on these datasets, we show that our LoL architectures are capable of processing low rank weight decompositions to predict CLIP score, finetuning data attributes, finetuning data membership, and accuracy on downstream tasks.

## 1 Introduction

Finetuning of pretrained models such as Large Language Models (Devlin et al., 2019; Brown et al., 2020; Dubey et al., 2024) and Diffusion models (Ho et al. (2020), Rombach et al. (2022)) for improved performance on tasks such as generating images in a specific style or creating text for mathematical proofs has become an extremely common paradigm in deep learning. While full finetuning of all weights effectively boosts model capabilities, it requires a large amount of memory and computation time. In recent years, Low Rank Adapation (LoRA) (Hu et al., 2021), a method for finetuning where a learnable low rank decomposition is added to each weight matrix ($W_i \mapsto W_i + U_i V_i^\top$), has been used as an alternative to other finetuning methods thanks to its increased efficiency. LoRA finetuning and its variants have become widespread; there are now software packages (Mangrulkar et al., 2022; Han & Han, 2023), paid services (Replicate, 2024), and online communities (Civitai, 2024) in which countless LoRAs are trained and shared.

Given the ubiquity of LoRA weights, one can imagine treating them as a data type. As in recent works in the emerging field of weight-space learning, we can process the weights of input neural networks by using other neural networks (often termed metanetworks (Lim et al., 2023a), neural functionals (Zhou et al., 2024b), or deep weight space networks (Navon et al., 2023a)).

In this work, we are the first to extensively study applications, theory and architectures for Learning on LoRAs (LoL), which encompasses any tasks where LoRA weights are the input to some predictive model. An LoL model acting on a LoRA could predict various useful properties of the underlying finetuned model. For instance, given LoRA weights of a finetuned model, an LoL model could predict the downstream accuracy of the model on some task, predict properties of the (potentially private) training data used to finetune the model, or edit the LoRA to work in some other

setting. Recently, Salama et al. (2024) and Dravid et al. (2024) also consider learning tasks on LoRA weights, for some specific applications and a few types of models or learning algorithms.

When developing LoL architectures for processing LoRA weights, we account for the structure and symmetries of this unique data type. For one, the low rank decomposition $(U, V)$ generally has significantly fewer parameters ($(n + m)r$) than the dense matrix $UV^\top$ ($nm$). This can be leveraged to more efficiently learn tasks on LoRA weights. Moreover, for any invertible matrix $R \in \text{GL}(r)$, we have that $URR^{-1}V^\top = UV^\top$, so the low rank decompositions $(U, V)$ and $(UR, VR^{-\top})$ are functionally equivalent. Because this transformation $\tau_R$ does not affect the underlying function represented by each LoRA, almost all relevant LoL problems are invariant to $\tau_R$. Therefore, the outputs of an effective model for any such task should be invariant to $\tau$.

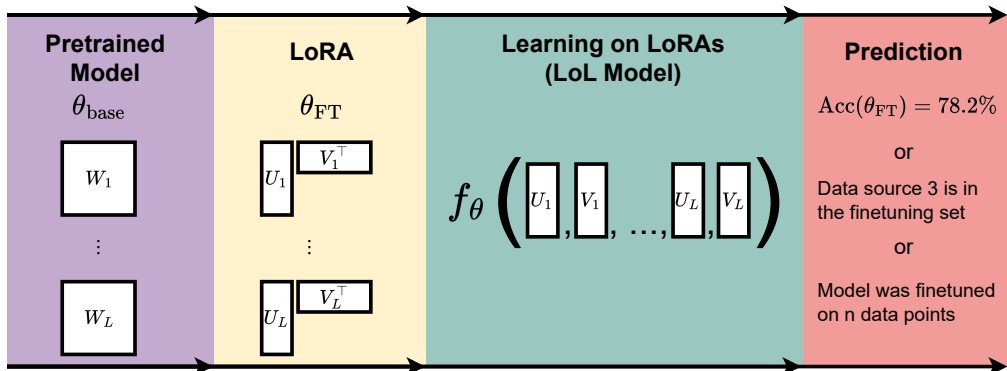

Figure 1: Overview of Learning on LoRAs (LoL). A pretrained model $\theta_{\text{base}}$ is finetuned to yield LoRA weight matrices $U_1, V_1, \ldots, U_L, V_L$. These LoRA weights are taken as input to an LoL model $f_\theta$, which can make predictions such as the downstream accuracy of the finetuned model.

To this end, motivated by previous successes in invariant and equivariant weight space learning (Navon et al., 2023a; Kofinas et al., 2024; Lim et al., 2023a; Zhou et al., 2024a; Kalogeropoulos et al., 2024) (and geometric deep learning in general (Bronstein et al., 2021)), we propose several $\text{GL}(r)$ equivariant and invariant neural architectures that can effectively process weights of LoRAs. Using various techniques from geometric deep learning such as canonicalization, invariant featurization, and equivariant linear maps, we develop several LoL models with various trade-offs in terms of efficiency, expressivity, and generalization.

To explore the feasibility of Learning on LoRAs tasks, and to analyze the effectiveness of our proposed architectures, we conduct experiments across various finetuned models. First, we create novel datasets for Learning on LoRAs; we train thousands of diverse LoRAs, which are finetuned from text-to-image diffusion generative models and language models. We then train LoL models on these LoRAs for prediction tasks such as: predict the CLIP score (Hessel et al., 2022) of a Stable Diffusion finetune given only its LoRA weights, predict attributes such as facial hair presence of the person that a diffusion model was personalized to, predict which data sources a language model LoRA was finetuned on, and predict the downstream reasoning accuracy of a language model LoRA. We find that simple $\text{GL}(r)$-invariant LoL models can often perform some of these tasks well, and equivariant-layer based LoL models can do well across most tasks.

## 2 BACKGROUND AND RELATED WORK

**LoRA background and symmetries** Hu et al. (2021) introduced the LoRA method, which is a parameter-efficient method for finetuning (typically large) models (Mangrulkar et al., 2022; Han et al., 2024). Consider a weight matrix $W \in \mathbb{R}^{n \times m}$ of some pretrained neural network. Directly finetuning $W$ would require training $nm$ parameters on additional data, which could be quite expensive. LoRA instead trains low-rank matrices $U \in \mathbb{R}^{n \times r}$ and $V \in \mathbb{R}^{m \times r}$ so that the new finetuned weight matrix is given by $W + UV^\top$. This only requires tuning $(n + m)r$ parameters, which is efficient as the rank $r$ is taken to be significantly lower than $n$ or $m$ (for instance a rank of $r = 8$ can be sufficient to finetune a 7B-parameter language model with $n = m = 4096$ (Liu et al., 2024)).

As mentioned in the introduction, for any invertible matrix $R \in \mathrm{GL}(r)$, $W + (UR)(VR^{-\top})^{\top} = W + UV^{\top}$, so the LoRA update given by $(UR, VR^{-\top})$ is functionally equivalent to the one given by $(U, V)$. As a special case, when $Q \in \mathrm{O}(r)$ is orthogonal, $(UQ, VQ)$ is also functionally equivalent. Many variants of the original LoRA work have been proposed, and they often have comparable symmetries that can be handled similarly in our framework (Mangrulkar et al., 2022; Han et al., 2024); we discuss LoRA variants and their symmetries in Appendix E.

Prior works have also studied continuous symmetries (Thomas et al., 2018; Bogatskiy et al., 2020; Satorras et al., 2022; Lim et al., 2023b; Pearce-Crump, 2023; Lawrence & Harris, 2024) such as rotation and scaling symmetries, especially for applications in the chemical and physical sciences. However, they study different classes of symmetries, such as $\mathrm{O}(n)$, $\mathrm{E}(n)$, and $\mathrm{SP}(n)$, none of which contain $\mathrm{GL}(n)$. To the best of our knowledge, we are the first to study $\mathrm{GL}(n)$ equivariant and invariant architectures.

**Weight-space learning and metanetworks** Several neural network architectures have been developed that take in neural network weights as input (Unterthiner et al., 2020; Eilertsen et al., 2020; Metz et al., 2022; Peebles et al., 2022; Navon et al., 2023a;b; Andreis et al., 2023; Shamsian et al., 2024). In many tasks, equivariance or invariance to parameter transformations that leave the network functionally unchanged has been found to be useful for empirical performance of weight-space networks (Navon et al., 2023a; Zhou et al., 2024b; Lim et al., 2023a; Kalogeropoulos et al., 2024; Tran et al., 2024). However, these works are not specialized for LoRA weight spaces, since they consider different symmetry groups: many such works focus only on discrete permutation symmetries (Navon et al., 2023a; Zhou et al., 2024b; Lim et al., 2023a; Zhou et al., 2024a), or scaling symmetries induced by nonlinearities (Kalogeropoulos et al., 2024; Tran et al., 2024). As covered in the previous section, LoRA weights have general invertible symmetries, which include as special cases certain permutations and scaling symmetries between $U$ and $V^{\top}$. While LoRAs contain inner permutation symmetries of the form $UP^{T}PV^{T} = UV^{\top}$, we emphasize that they generally do not have outer permutation symmetries of the more recognizable form $P_1 UV^{T} P_2$, which would resemble those of standard weight spaces (e.g. MLPs or CNNs) more. This is an important difference; for instance, the outer permutation symmetries significantly harm linear merging of models trained from scratch (Entezari et al., 2022; Ainsworth et al., 2023; Lim et al., 2024), whereas finetuned models can be merged well with simple methods (Wortsman et al., 2022; Ilharco et al., 2023).

Recently, two works have explored the weight space of LoRA finetuned models for certain tasks. Salama et al. (2024) develop an attack that, given the LoRA weights of a finetuned model, predicts the size of the dataset used to finetune the model. Inspired by correlations between finetuning dataset size and singular value magnitudes, they define a very specific type of LoL model that takes the singular values of dense, multiplied-out LoRA weights $\sigma_1(U_i V_i^{\top}), \ldots, \sigma_r(U_i V_i^{\top})$ as input. In contrast, we study more tasks, consider the problem of general LoL model design, and develop more expressive and efficient LoL models in our paper. In another context, Dravid et al. (2024) study the LoRA weight space of finetuned personalized text-to-image diffusion models. They do not define LoL models, but instead study operations such as linear edits in the principal component space of their rank-one LoRA weights.

# 3 LEARNING ON LORAS ARCHITECTURES

Table 1: Properties of LoL architectures that we propose in this paper. Runtimes are for one LoRA layer with $U \in \mathbb{R}^{n \times r}$ and $V \in \mathbb{R}^{m \times r}$, so the rank $r$ is generally much lower than $n$ and $m$. Expressivity refers specifically to a notion of ability to fit GL-invariant functions, which is formalized in Definition C.1.

| Model | GL-Invariant | O-Invariant | Expressive | Preprocess Time | Forward Time |
|---|---|---|---|---|---|
| MLP | ✗ | ✗ | ✓ | $O((m+n)r)$ | $O((m+n)r)$ |
| MLP + O-Align | ✗ | ✓ | ✓ | $O((m+n)r^2)$ | $O((m+n)r)$ |
| MLP + SVD | ✓ | ✓ | ✗ | $O((m+n)r^2)$ | $O((m+n)r)$ |
| MLP + Dense | ✓ | ✓ | ✓ | $O(mnr)$ | $O(mn)$ |
| GL-net | ✓ | ✓ | ✓ | $O((m+n)r)$ | $O((m+n)r)$ |

In this section, we develop several neural network architectures for Learning on LoRAs. These have different trade-offs and properties, which we summarize in Table 1. First, we mathematically formalize the LoL learning problem. Then we describe four methods based on feeding features into a simple Multi-layer Perceptron (MLP). [1] Finally, we derive GL-net, an architecture based on various equivariant and invariant modules.

## 3.1 LEARNING ON LoRAs MODELS

We define a Learning on LoRAs (LoL) model that takes LoRA updates as input as follows. Suppose there are $L$ matrices in a base model that are finetuned via LoRA, i.e. the LoRA weights are $(U_1, V_1), \ldots, (U_L, V_L)$, where $U_i \in \mathbb{R}^{n_i \times r}$ and $V_i \in \mathbb{R}^{m_i \times r}$. An LoL model with parameters $\theta$ and output space $\mathcal{Y}$ is a function $f_\theta : \mathbb{R}^{\sum_i n_i r + \sum_i m_i r} \to \mathcal{Y}$. An LoL model can output a scalar prediction ($\mathcal{Y} = \mathbb{R}$) or latent LoRA weight representations $(\tilde{U}_1, \tilde{V}_1), \ldots, (\tilde{U}_L, \tilde{V}_L)$ where $\tilde{U}_i \in \mathbb{R}^{\tilde{n}_i \times r}$ and $\tilde{V}_i \in \mathbb{R}^{\tilde{m}_i \times r}$ ($\mathcal{Y} = \mathbb{R}^{\sum_i \tilde{n}_i r + \sum_i \tilde{m}_i r}$).

An LoL model is called GL-*invariant* if for all $R_1, \ldots, R_L \in \mathrm{GL}(r)$

$$f_\theta(U_1 R_1, V_1 R_1^{-\top}, \ldots, U_L R_L, V_L R_L^{-\top}) = f_\theta(U_1, V_1, \ldots, U_L, V_L). \tag{1}$$

This represents invariance to the action of the direct product $\mathrm{GL}(r) \times \cdots \times \mathrm{GL}(r)$ of $\mathrm{GL}(r)$ with itself $L$-times ($\mathrm{GL}(r)^L$). When the output space has decomposition structure, i.e. $f_\theta(U_1, V_1, \ldots, U_L, V_L) = ((\tilde{U}_1, \tilde{V}_1), \ldots, (\tilde{U}_L, \tilde{V}_L))$, we say that the model is $\mathrm{GL}(r)$-equivariant if for all $R_1, \ldots, R_L \in \mathrm{GL}(r)$,

$$f_\theta(U_1 R_1, V_1 R_1^{-\top}, \ldots, U_L R_L, V_L R_L^{-\top}) = ((\tilde{U}_1 R_1, \tilde{V}_1 R_1^{-\top}), \ldots, (\tilde{U}_L R_L, \tilde{V}_L R_L^{-\top})). \tag{2}$$

Expressivity is an important property of LoL models. Informally, an LoL model is universally expressive if it can fit any nice GL-invariant function to arbitrary accuracy. We give a formal definition in Definition C.1, and prove our results in the context of this formal definition.

## 3.2 MLP-BASED METHODS WITH FEATURIZATION

**MLP: Simple MLP on LoRA Weights**  The simplest of our models is a simple MLP that takes the flattened LoRA weights as input: $\mathrm{MLP}_\theta(U_1, V_1, \ldots, U_L, V_L)$. This method is not invariant to the LoRA parameter symmetries, but it is fully expressive, and it is efficient in that it does not need to form the $n$-by-$m$ dense matrices $UV^\top$.

**MLP + O-Align: Alignment for Canonicalization**  One method for designing invariant or equivariant networks is to canonicalize the input by using a symmetry-invariant transformation to convert it into a canonical form (Kaba et al., 2023; Dym et al., 2024; Ma et al., 2024). For example, to canonicalize a dataset of point clouds with respect to rotations, one might choose one specific point cloud and then rotate all other point clouds in the dataset to it to maximize their relative similarity. After this alignment, any standard architecture can process the point clouds in an invariant manner without taking rotation into account. This kind of rotation alignment, known as $\mathrm{O}(r)$ canonicalization, admits a closed from solution and is significantly simpler than $\mathrm{GL}(r)$ canonicalization.

We canonicalize LoRA weights $U_i, V_i$ into $\mathrm{O}(r)$-invariant representatives $U_i Q_i, V_i Q_i$ as follows. First, we select template weights $\mathbf{U}_i, \mathbf{V}_i$ of the same shape as $U_i$ and $V_i$ (in our experiments we randomly select $\mathbf{U}_i$ and $\mathbf{V}_i$ from our training set of LoRAs). Then we align $U_i, V_i$ to the templates by finding the orthogonal matrix $Q_i$s that most closely match them in the Frobenius norm:

$$Q_i = \underset{Q_i \in \mathrm{O}(r)}{\arg\min} \|U_i Q_i - \mathbf{U}_i\|_F^2 + \|V_i Q_i - \mathbf{V}_i\|_F^2. \tag{3}$$

This is an instance of the well-known Orthogonal Procrustes problem (Schönemann, 1966), which has a closed form solution: $Q_i$ can be computed from the singular value decomposition of the $r$-by-$r$ matrix $U_i^\top \mathbf{U}_i + V_i^\top \mathbf{V}_i$, which can be computed efficiently when the rank $r$ is low. After computing these $Q_i$s for pair of LoRA weights, we input the $U_i Q_i, V_i Q_i$ into an MLP to compute predictions.

---

[1] These features can also be processed by other generic architectures, such as recurrent or Transformer-based sequence models along the layer dimension, but we use only MLPs in the work for simplicity.

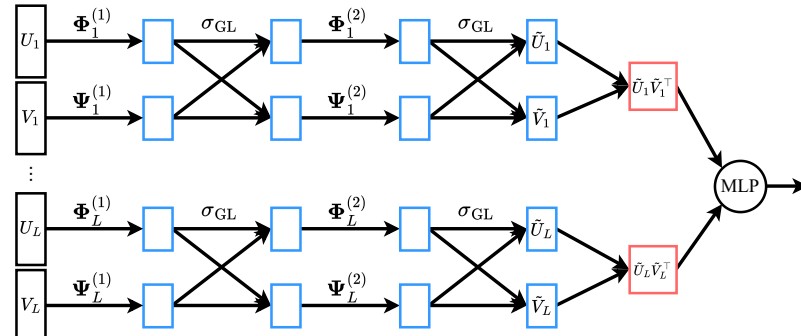

Figure 2: Architecture of GL-net. Blue boxes are equivariant representations, and red boxes are invariant representations. First, equivariant linear maps lower the dimension of the input. Then our GL equivariant nonlinearities and more equivariant linear maps process the features. Finally, a matrix multiplication head computes invariant features that are processed by an MLP.

**MLP + SVD: Singular Values as Features**  Similarly to Salama et al. (2024), we also consider an LoL architecture that feeds the singular values of the multiplied-out LoRA weights, $\sigma_1(U_i V_i^\top), \ldots, \sigma_r(U_i V_i^\top)$ into an MLP to compute predictions (though Salama et al. (2024) mostly use nearest neighbor predictors instead of an MLP). Since $U_i V_i^\top$ is rank $r$, there are at most $r$ nonzero singular values. These singular values are GL-invariant features, but they are not expressive. For instance, negating $U_i$ does not affect the singular values, but it can completely change the functionality and destroy the performance of the finetuned model.

**MLP + Dense: Multiplying-Out LoRA Weights**  Lastly, a simple, natural method is to first perform matrix multiplications to compute the full dense matrices $U_i V_i^\top$, and then apply a machine learning model to these dense matrices. In this paper, we will flatten and concatenate these dense matrices, and then apply a simple MLP to them. This approach is fully GL-invariant, and is universally expressive, but it is computationally expensive since the dense matrices are of size $nm$ as opposed to the size $(n + m)r$ of the low rank decomposition.

### 3.3 GL-NET: CONSTRUCTING G-INVARIANT MODELS USING EQUIVARIANT LAYERS

A common and effective method for parameterizing invariant neural networks is to first process the input with equivariant layers and then make the final prediction with invariant modules (Cohen & Welling, 2016; Maron et al., 2019a;b; Bronstein et al., 2021). Thus, we develop GL-net, which consists of a series of GL-equivariant linear layers and equivariant nonlinearities followed by a GL-invariant head to predict invariant characteristics of the input. See Figure 2 for an illustration.

#### 3.3.1 EQUIVARIANT LINEAR LAYERS

For LoRA weights $U_i \in \mathbb{R}^{n_i \times r}$ and $V_i \in \mathbb{R}^{m_i \times r}$, we describe the GL-equivariant linear layers, which map to new weights $\tilde{U}_i \in \mathbb{R}^{n_i' \times r}$ and $\tilde{V}_i \in \mathbb{R}^{m_i' \times r}$. For each of these LoRA weights, we have LoL model parameters $\mathbf{\Phi}_i \in \mathbb{R}^{n_i' \times n_i}$ and $\mathbf{\Psi}_i \in \mathbb{R}^{m_i' \times m_i}$. The equivariant linear map is given by:

$$F_{\text{Linear}}(U_1, V_1, \ldots, U_L, V_L) = (\mathbf{\Phi}_1 U_1, \mathbf{\Psi}_1 V_1, \ldots, \mathbf{\Phi}_L U_L, \mathbf{\Psi}_L V_L) \tag{4}$$

That is, a GL equivariant linear layer consists of left matrix multiplying each $U_i$ by a learnable $\mathbf{\Phi}_i$, and left matrix multiplying each $V_i$ by a learnable $\mathbf{\Psi}_i$. In practice, we choose the same hidden dimension for every $\tilde{U}_i$ and $\tilde{V}_i$ for simplicity, so $n_1' = m_1' = \ldots = n_L' = m_L'$. The equivariance of $F_{\text{Linear}}$ can be easily seen. For instance, if there is only one layer ($L = 1$), we have

$$F_{\text{Linear}}(U_1 R, V_1 R^{-\top}) = (\mathbf{\Phi}_1 U_1 R, \mathbf{\Psi}_1 V_1 R^{-\top}) = (\tilde{U}_1 R, \tilde{V}_1 R^{-\top}) = R \star F_{\text{Linear}}(U_1, V_1), \tag{5}$$

where $(\tilde{U}_1, \tilde{V}_1) = F_{\text{Linear}}(U_1, V_1)$, and $R \star (\tilde{U}_1, \tilde{V}_1) = (\tilde{U}_1 R, \tilde{V}_1 R^{-\top})$ is the action of $R$ on the LoRA space. In fact, we can show a stronger statement — these linear maps in equation 4 constitute *all possible* GL-equivariant linear maps. See Appendix A for the proof.

**Proposition 1.** *All linear GL-equivariant layers can be written in the form of equation 4.*

**Extension to Convolution LoRAs.** Low rank convolution decompositions are generally represented as two consecutive convolutions where $C_B$ projects the input down from $m$ to $r$ channels and $C_A$ projects the input up to $n$. For our architecture we flatten all dimensions of the convolutions except the hidden channel dimension, and then we apply equivariant linear layers.

### 3.3.2 EQUIVARIANT NON-LINEARITY

Equivariant networks often interleave pointwise non-linearities with linear equivariant layers. Unfortunately, non-trivial pointwise non-linearities are not equivariant to our symmetry group. A general recipe for designing equivariant non-linearities is taking $f_{\text{equi}}(\mathbf{x}) = f_{\text{inv}}(\mathbf{x}) \cdot \mathbf{x}$, for some scalar invariant function $f_{\text{inv}}$ (Thomas et al., 2018; Villar et al., 2021; Blum-Smith & Villar, 2022). In our case, given any non-linearity $\sigma : \mathbb{R} \to \mathbb{R}$, we define a GL-equivariant non-linearity by

$$\sigma_{\text{GL}}(U)_i = \sigma\Big(\sum_j (UV^\top)_{ij}\Big)U_i, \qquad \sigma_{\text{GL}}(V)_i = \sigma\Big(\sum_j (UV^\top)_{ji}\Big)V_i. \tag{6}$$

In other words, we scale the $i$-th row of $U_i$ by a quantity that depends on the the $i$-th row sum of $UV^\top$, and scale the $i$-th row of $V_i$ by a quantity that depends on the $i$-th column sum of $UV^\top$. Since $UV^\top$ is invariant under the action of GL, and GL acts independently on each row of $U$ and $V$, $\sigma_{\text{GL}}$ is equivariant (we prove this in Appendix B). In our experiments, we often take $\sigma(x) = \text{ReLU}(\text{sign}(x))$, which has the effect of zero-ing out entire rows of $U$ or $V$ in a GL-equivariant way — this is a natural GL-equivariant generalization of ReLU. For our experiments we tune the number of equivariant linear layers and we frequently find that only one is required, so we often do not use this equivariant non-linearity; nonetheless, it may be more useful in other applications, such as GL-equivariant tasks.

### 3.3.3 INVARIANT HEAD

Many equivariant architectures used for classification use an invariant aggregation step before applying a standard classifier model. For invariant classification tasks, we use an invariant head which consists of the relatively simple operation of computing the LoRA matrix products, concatenating them, and then applying an MLP on top. Explicitly, this is given as $f_{\text{inv}}(U_1, V_1, \ldots, U_L, V_L) = \text{MLP}(\text{cat}[U_1 V_1^\top, \ldots, U_L V_L^\top])$, where $\text{cat}$ concatenates the entries of the inputs into a flattened vector. For the input, computing $U_i V_i^T$ would be both memory and time expensive. To avoid this, we use equivariant linear layers to lower the dimension of $U_i, V_i$ to about $32 \times r$, such that $U_i V_i^\top \in \mathbb{R}^{32 \times 32}$ is efficient to compute – see Figure 2 for an illustration.

## 4 THEORY

Here, we restate the properties of our models as described in Section 3 and Table 1. All proofs of these results are in the Appendix.

**Theorem 1** (Invariance). *MLP + Dense, MLP + SVD,* GL*-net are* GL*-Invariant. MLP +* O*-Align is* O*-Invariant but not* GL*-Invariant. MLP is not* GL *or* O*-Invariant.*

**Theorem 2** (Universality). *MLP, MLP +* O*-Align, MLP + Dense, and* GL*-net can arbitrarily approximate any GL-invariant continuous function on a compact set of full rank matrices.*

## 5 EXPERIMENTAL RESULTS

In this section, we present experimental results evaluating our five LoL models across tasks involving finetuned diffusion and language models (Subsections 5.2- 5.3). Our experiments demonstrate the efficacy of LoL models in solving GL-invariant tasks on LoRAs. Additionally, we explore these models' ability to generalize to LoRAs of previously unseen ranks, with detailed findings presented in Subsection 5.4.

### 5.1 DATASETS OF TRAINED LoRA WEIGHTS

We generate three new datasets of LoRAs of varying ranks. Two of these datasets correspond to finetunes of diffusion models on image datasets, while the last corresponds to finetunes of a language

model on text. Our datasets have diversity in terms of LoRA rank, training hyperparameters (two datasets have randomly sampled hyperparameters, whereas one has fixed hyperparameters), and base model architecture.

**CelebA-LoRA.** We train a dataset of 3,900 LoRA finetuned Stable Diffusion 1.4 models (Rombach et al., 2022) using the PEFT library (Mangrulkar et al., 2022). Each LoRA is rank 4, and is finetuned via DreamBooth personalization (Ruiz et al., 2023) on 21 images of a given celebrity in the CelebA dataset (Liu et al., 2015). Further, each LoRA is trained with randomly sampled hyperparameters (gradient accumulation steps, train steps, learning rate, prompt) and initialization. We train LoL models to predict hyperparameters, CLIP scores, and training images of finetuned diffusion models by using their LoRA weights.

**Imagenette-LoRA.** We also use Dreambooth to finetune another 2,046 Stable Diffusion 1.4 models with LoRA rank 32 on different subsets of the Imagenette dataset (Howard, 2019) — a subset of ImageNet (Deng et al., 2009) consisting of images from ten dissimilar classes. Unlike CelebA LoRA, we use the same training hyperparameters for each finetuning run (but vary initialization, random seed, and finetuning dataset).

**Qwen2-ARC-LoRA.** We create a dataset of 2,000 language model LoRAs by finetuning the 1.5 billion parameter Qwen2 model (Yang et al., 2024) on subsets of the training set of the commonly used ARC dataset (Clark et al., 2018), which is a dataset for testing question answering and science knowledge. The ARC dataset consists of questions from many data sources. For each LoRA, we randomly sample a subset of 19 data sources, and omit data from the unsampled data sources. Also, each LoRA is randomly initialized and trained with randomly sampled hyperparameters. See Appendix D.2 for more details.

## 5.2 DIFFUSION MODEL CLASSIFICATION

Table 2: We report train and test mean squared error (MSE) for predicting normalized CLIP score of CelebA-LoRA models. We also show test Kendall's $\tau$ and $R^2$ coefficients. GL-net has significantly lower test MSE than any other LoL model, whereas a standard MLP does no better than random guessing. MLPs with O-Align, SVD, or Dense featurization all perform similarly on the test set.

| LoL Model | Train MSE | Test MSE | $\tau$ | $R^2$ |
|---|---|---|---|---|
| MLP | $.047 \pm .004$ | $.988 \pm .005$ | $.226 \pm .016$ | $.333 \pm .022$ |
| MLP + O-Align | $.001 \pm .001$ | $.175 \pm .006$ | $.654 \pm .004$ | $.856 \pm .004$ |
| MLP + SVD | $.071 \pm .003$ | $.148 \pm .005$ | $.667 \pm .008$ | $.863 \pm .006$ |
| MLP + Dense | $.013 \pm .002$ | $.169 \pm .011$ | $.677 \pm .006$ | $.871 \pm .005$ |
| GL-net | $.009 \pm .001$ | $\mathbf{.111 \pm .004}$ | $\mathbf{.695 \pm .005}$ | $\mathbf{.884 \pm .003}$ |

### 5.2.1 CLIP SCORE PREDICTION.

CLIP scores were introduced by Hessel et al. (2022) as an automated method for measuring text-image-alignment in diffusion models by evaluating the semantic similarity between their prompts and generated images. Higher CLIP scores generally correspond to better diffusion models, so CLIP score is a useful metric for evaluating these models. We calculate the CLIP score of each of our 3,900 diffusion models (with 33 fixed prompts) and train LoL models on the task of determining the CLIP score of a model given its finetuned weights; see Appendix D.1.1 for more details.

The results are shown in Table 2. All of the LoL models (besides the vanilla MLP) can effectively predict the CLIP score of diffusion LoRAs using only their decomposed low rank matrices. The poor performance of vanilla MLP demonstrates the importance of GL symmetries, whereas the relatively good performance of MLP + Dense suggests "outer" permutation symmetries are less important for LoL models, confirming two of our hypotheses from Section 2. GL-net is able to calculate CLIP score significantly faster than generating images, which requires 660 score-network forward passes per finetuned model. Other baselines, including MLP + SVD, MLP + O-Align and MLP + Dense are less predictive. To demonstrate the utility of this task, we generate images from the two models in

Figure 3: Images generated by two diffusion models in our test set. (a), (c), and (e) correspond to images generated by the model predicted by GL-net to have the highest CLIP score. (b), (d), and (f) correspond to outputs of the model predicted by GL-net to have the lowest CLIP score.

| (a) Best Model | (b) Worst Model | (c) Best Model | (d) Worst Model | (e) Best Model | (f) Worst Model |

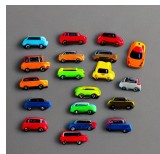 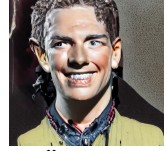 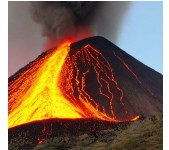 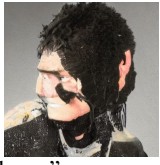 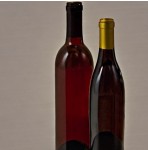 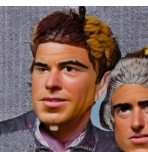

"toy cars" "a volcano" "two wine bottles"

Table 3: Results for using LoL models to predict properties of the finetuning data of diffusion models, given only the LoRA weights. In the left section, the task is to predict 5 different binary attributes of the CelebA celebrity that each LoRA was finetuned on. In the right section, the task is to predict which of the 10 Imagenette classes were included in the finetuning data of the LoRA. MLP and MLP + SVD have trouble fitting these tasks, while GL-net performs the best overall.

|  | CelebA Attributes | | | Imagenette Classes | | |
| --- | --- | --- | --- | --- | --- | --- |
| LoL Model | Train Loss | Test Loss | Test Acc | Train Loss | Test Loss | Test Acc |
| MLP | .551 ± .000 | .554 ± .000 | 72.4 ± 0.0 | .582 ± .004 | .709 ± .004 | 49.6 ± 1.3 |
| MLP + O-Align | .074 ± .022 | .333 ± .008 | 87.2 ± 0.5 | .002 ± .001 | .278 ± .008 | 87.8 ± 0.3 |
| MLP + SVD | .490 ± .015 | .509 ± .013 | 77.3 ± 1.3 | .581 ± .013 | .638 ± .013 | 65.6 ± 0.6 |
| MLP + Dense | .131 ± .018 | .267 ± .007 | 89.1 ± 0.4 | .019 ± .001 | .264 ± .011 | 88.9 ± 0.6 |
| GL-net | .064 ± .008 | **.232 ± .007** | **91.3 ± 0.1** | .019 ± .000 | **.244 ± .005** | **90.4 ± 0.3** |

our test set predicted by GL-net to have the highest and lowest CLIP scores respectively. As shown in Figure 3, GL-net's predictions are highly correlated with the actual quality of model output.

### 5.2.2 PREDICTING PROPERTIES OF TRAINING DATA

**Dataset attribute prediction.** On our CelebA-LoRA dataset, we train LoL models to classify attributes of the celebrity that each LoRA was finetuned on. Due to the noisy nature of CelebA labels, we follow the advice of (Lingenfelter et al., 2022) and only train and test on the five CelebA attributes they determine to be least noisy. Also, we train LoL models to predict which Imagenette classes appeared in the finetuning data of each model in Imagenette-LoRA. Our results are in Table 3.

**Dataset size prediction.** The task of predicting the size of the finetuning dataset given LoRA weights was first studied recently by Salama et al. (2024). The success of this task implies a privacy leak, since model developers may sometimes wish to keep the size of their finetuning dataset private. Moreover, the dataset size is a useful quantity to know for data membership inference attacks and model inversion attacks, so accurately predicting dataset size could improve effectiveness of these attacks too (Shokri et al., 2017; Haim et al., 2022).

In Table 4, we show results for finetuning-dataset-size prediction using LoL models. The task is to take in the LoRA weights of one of the diffusion

Table 4: Results for finetuning dataset size prediction with LoL models. We use our Imagenette-LoRA dataset, and predict the number of classes (or equivalently images) that are used to finetune each model.

| LoL Model | Train Loss | Test Acc |
| --- | --- | --- |
| MLP | .439 ± .030 | 20.9 ± 3.1 |
| MLP + O-Align | .014 ± .004 | 35.2 ± 2.1 |
| MLP + SVD | .082 ± .010 | **73.1 ± 1.6** |
| MLP + Dense | .382 ± .119 | 31.6 ± 4.3 |
| GL-net | .005 ± .003 | 46.8 ± 2.1 |

models from our Imagenette-LoRA dataset, and predict the number of unique images that it was finetuned on. Although it struggles on some other tasks, we see that MLP + SVD is able to effectively predict dataset size, in line with observations from Salama et al. (2024). Other LoL models struggle to generalize well on this task. Interestingly, GL-net performs approximately as well as

expected if using the following strategy: first predict which classes are present in the finetuning set of the LoRA (as in Table 3), and then sum the number of classes present to predict the dataset size. MLP+SVD is clearly using different predictive strategies for this task, as it cannot predict which individual classes are present (Table 3), but it can predict the number of classes present. See Appendix D.3 for more analysis on the learned prediction strategies of the LoL models on this task.

## 5.3 Language Model Classification

Table 5: LoL model performance on LoRAs of the Qwen2 1.5B language model. The left column is prediction of which data sources the input LoRA was finetuned on, the middle column is prediction of the validation loss for the finetuning task, and the right column is prediction of the accuracy of the LoRA on the ARC-C (Clark et al., 2018) test set. All metrics are reported on the LoL task's test set (on held-out input networks). Higher numbers are better on all metrics.

| | LoL Model Prediction Target | | |
|---|---|---|---|
| LoL Model | Data Membership (Acc) | Val Loss ($R^2$) | ARC-C Acc ($R^2$) |
| MLP | $.516 \pm .006$ | $.113 \pm .059$ | $.107 \pm .035$ |
| MLP + O-Align | $.550 \pm .016$ | $.821 \pm .078$ | $.965 \pm .004$ |
| MLP + SVD | $.551 \pm .001$ | $\mathbf{.999} \pm .000$ | $.983 \pm .002$ |
| MLP + Dense | $\mathbf{.625} \pm .008$ | $.987 \pm .003$ | $.981 \pm .002$ |
| GL-net | $.605 \pm .007$ | $.998 \pm .000$ | $\mathbf{.987} \pm .001$ |

In this section, we experiment with LoL models on our Qwen2-ARC-LoRA dataset of finetuned language models. For our first task, we consider a type of data membership inference task: we aim to predict whether each of the 19 data sources was used to train a given LoRA (this is a 19-label binary classification task). We also consider two performance prediction task: we train LoL models to predict, for each LoRA, the validation loss on the finetuning objective, and the downstream accuracy on the ARC-C test set (Clark et al., 2018).

Results are in Table 5. Several of our LoL models are very successful at predicting the finetuning validation loss and ARC-C test accuracy of LoRA-finetuned language models, with some of them achieving .99 $R^2$ on finetuning validation loss regression and over .98 $R^2$ on ARC-C test accuracy regression. This could be useful in evaluations of finetuned models, as standard evaluations of language models can require a lot of time and resources, whereas our LoL models can evaluate these models with one quick forward pass. However, data membership inference is a harder task for the LoL models, with the best model achieving only 62.5% test accuracy in predicting which data sources were present in the finetuning dataset. Though our setup is quite different, these results could be related to work showing that model-based membership inference attacks are challenging for LLMs (Duan et al., 2024; Das et al., 2024).

## 5.4 Generalization to Unseen Ranks

In the previous sections, each LoL model was trained on LoRA weights of one fixed rank (e.g. rank 4 for the LLMs, rank 32 for the Imagenette models). In practice, model developers can choose different LoRA ranks for finetuning their models, even when they are finetuning the same base model for similar tasks. Thus, we may desire LoL models that are effective for inputs of different ranks. In this section, we explore an even more difficult problem — whether LoL models can generalize to ranks that are unseen during training time.

The MLP + Dense and GL-net models can directly take as input models of different ranks. We also use MLP + SVD on inputs of different ranks, by parameterizing the model for inputs of rank $r$, then truncating to top $r$ singular values for inputs of rank greater than $r$, and zero-padding to length $r$ for inputs of rank less than $r$.

In Figure 4, we train LoL models on inputs of rank 4, and then test performance on inputs of ranks between 1 and 32. On the CelebA attribute prediction task, we see that MLP + Dense and GL-net mostly generalize very well to different ranks that are unseen during training (except not as well for $r = 1$), while MLP + SVD does not generalize as well.

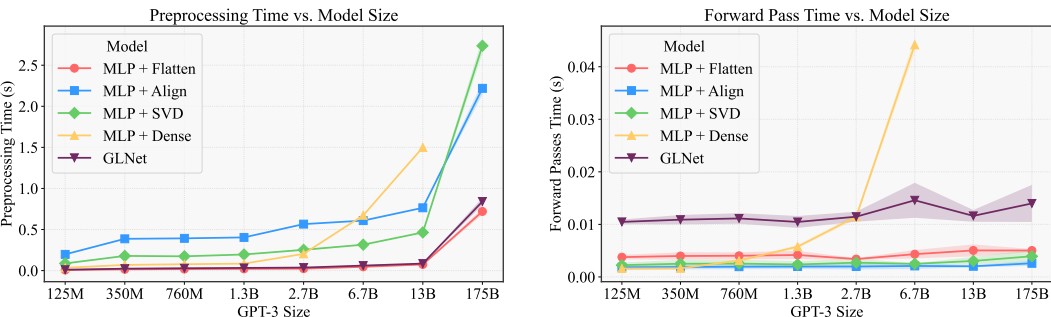

Figure 4: Performance of LoL models across inputs of varying ranks. Each model is only trained on rank 4 LoRA weights from CelebA-LoRAs. (Left) Test accuracy on CelebA attribute prediction. (Right) Test loss on CelebA attribute prediction. MLP + Dense and GL-net generalize well to ranks that are unseen during training, but do face degradation at rank one. On the other hand, MLP + SVD does not generalize well.

Figure 5: (Left) data preprocessing time for LoL models across 64 inputs of varying sizes. (Right) forward pass time for LoL models across 512 inputs of varying sizes. MLP + Dense runs out of memory for largest inputs.

## 5.5 Runtime and Scaling to Large Models

Previous weight-space models generally take in small networks as inputs, e.g. 5,000 parameters (Unterthiner et al., 2020), 80,000 parameters (Lim et al., 2023a), or sometimes up to 4 million parameters (Navon et al., 2023b). However, many of the most impactful neural networks have orders of magnitude more parameters. Thus, here we consider the runtime and scalability of LoL models as we increase the model size. We will consider all of the language model sizes in the GPT-3 family (Brown et al., 2020), which range from 125M to 175B parameters. We assume that we finetune rank 4 LoRA weights for one attention parameter matrix for each layer.

In Figure 5, we show the time it takes for data preprocessing of 64 input networks, and forward passes for 512 input networks (with batch size up to 64). Even at the largest scales, it only takes at most seconds to preprocess and compute LoL model forward passes for each input LoRA. Every LoL model besides MLP + Dense scales well with the model size: forward passes barely take longer at larger model sizes, and data preprocessing is still limited to less than a second per input network.

## 6 Conclusion

In this work, we introduced the Learning on LoRAs framework, and investigated architectures, theory, and applications for it. There are many potential applications and future directions for using LoL models to process finetunes. For instance, future work could explore equivariant tasks, such as those that involve editing or merging LoRAs. Additionally, future work could consider learning an LoL model that can generalize across different model architectures, or different base models of the same architecture.

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

## A  GL EQUIVARIANT LINEAR MAPS CHARACTERIZATION

Here, we characterize the form of GL-equivariant linear maps, and provide the proof of Proposition 1. We use basic representation theory techniques, are similar to the ones used to characterize equivariant linear maps in the geometric deep learning literature (Maron et al., 2019a; Finzi et al., 2021; Navon et al., 2023a).

### A.1  PROOF OF PROPOSITION 1

First, we prove a lemma, that allows us to analyze equivariant linear maps between direct sums of spaces in terms of a direct sum of equivariant linear maps between the constituent spaces. For a group $G$, we denote the vector space of $G$-equivariant linear maps from $\mathcal{V}$ to $\mathcal{W}$ by $\mathrm{Hom}_G(\mathcal{V}, \mathcal{W})$. This is closely related to a result from Navon et al. (2023b) that characterizes the matrices underlying linear maps between direct sums.

**Lemma 1.** *Let $G$ be a group and let $\mathcal{V}_1, \ldots, \mathcal{V}_n$, $\mathcal{W}_1, \ldots, \mathcal{W}_m$ be $G$-representations. If $\mathcal{V} = \mathcal{V}_1 \oplus \cdots \oplus \mathcal{V}_n$ and $\mathcal{W} = \mathcal{W}_1 \oplus \cdots \oplus \mathcal{W}_m$ are direct sums of representations then*

$$\mathrm{Hom}_G(\mathcal{V}, \mathcal{W}) \cong \bigoplus_{i,j} \mathrm{Hom}_G(\mathcal{V}_i, \mathcal{W}_j). \tag{7}$$

*Proof.* We construct an explicit isomorphism $F : \mathrm{Hom}_G(\mathcal{V}, \mathcal{W}) \to \bigoplus_{i,j} \mathrm{Hom}_G(\mathcal{V}_i, \mathcal{W}_j)$. Let $\mathrm{inc}_i : \mathcal{V}_i \to \mathcal{V}$ denote the inclusion of $\mathcal{V}_i$ in $\mathcal{V}$, and $\mathrm{proj}_j : \mathcal{W} \to \mathcal{W}_j$ denote the projection from $\mathcal{W}$ to $\mathcal{W}_j$. For $\phi \in \mathrm{Hom}_G(\mathcal{V}, \mathcal{W})$, define:

$$F(\phi) = (\mathrm{proj}_j \circ \phi \circ \mathrm{inc}_i)_{i,j} \tag{8}$$

$F$ is clearly linear, being defined by composition of linear maps. Moreover, since $\phi$, $\mathrm{inc}_i$, and $\mathrm{proj}_j$ are all $G$-equivariant, their composition $\phi_{i,j} = \mathrm{proj}_j \circ \phi \circ \mathrm{inc}_i$ is also $G$-equivariant. To prove injectivity, suppose $F(\phi) = F(\psi)$ for some $\phi, \psi \in \mathrm{Hom}_G(\mathcal{V}, \mathcal{W})$. Then $\forall v = (v_1, \ldots, v_n) \in \mathcal{V}$

$$\phi(v) = (\mathrm{proj}_1(\phi(v)), \ldots, \mathrm{proj}_m(\phi(v)))$$

$$= \left( \sum_i \mathrm{proj}_1(\phi(\mathrm{inc}_i(v_i))), \ldots, \sum_i \mathrm{proj}_m(\phi(\mathrm{inc}_i(v_i))) \right)$$

$$= \left( \sum_i \mathrm{proj}_1(\psi(\mathrm{inc}_i(v_i))), \ldots, \sum_i \mathrm{proj}_m(\psi(\mathrm{inc}_i(v_i))) \right)$$

$$= \psi(v).$$

For surjectivity, given $(\phi_{i,j})_{i,j} \in \bigoplus_{i,j} \mathrm{Hom}_G(\mathcal{V}_i, \mathcal{W}_j)$, define $\phi : \mathcal{V} \to \mathcal{W}$ by

$$\phi(v_1, \ldots, v_n) = \left( \sum_i \phi_{i,1}(v_i), \ldots, \sum_i \phi_{i,m}(v_i) \right). \tag{9}$$

$\phi$ is linear by construction; to show $G$-equivariance, let $g \in G$ and $(v_1, \ldots, v_n) \in \mathcal{V}$

$$\phi(g \cdot (v_1, \ldots, v_n)) = \phi(g \cdot v_1, \ldots, g \cdot v_n)$$

$$= \left( \sum_i \phi_{i,1}(g \cdot v_i), \ldots, \sum_i \phi_{i,m}(g \cdot v_i) \right)$$

$$= \left( \sum_i g \cdot \phi_{i,1}(v_i), \ldots, \sum_i g \cdot \phi_{i,m}(v_i) \right)$$

$$= g \cdot \phi(v_1, \ldots, v_n),$$

where we use the $G$-equivariance of each $\phi_{i,j}$. Thus, $\phi \in \mathrm{Hom}_G(\mathcal{V}, \mathcal{W})$, and by construction $F(\phi) = (\phi_{i,j})_{i,j}$. Therefore, $F$ is a linear bijection. □

*Proof of Proposition 1.* Let $\mathcal{I} := (\mathcal{U}_1 \oplus \mathcal{V}_1^*) \oplus \cdots \oplus (\mathcal{U}_L \oplus \mathcal{V}_L^*)$ and $\mathcal{O} := (\tilde{\mathcal{U}}_1 \oplus \tilde{\mathcal{V}}_1^*) \oplus \cdots \oplus (\tilde{\mathcal{U}}_L \oplus \tilde{\mathcal{V}}_L^*)$ be the input and output spaces of an equivariant LoL model. Denote $\dim(\mathcal{U}_i) = n_i \cdot r$, $\dim(\mathcal{V}_i^*) = r \cdot m_i$, $\dim(\tilde{\mathcal{U}}_i) = n_i' \cdot r$, $\dim(\tilde{\mathcal{V}}_i^*) = r \cdot m_i'$, and $G := \mathrm{GL}(r)^L = \mathrm{GL}(r) \times \cdots \times \mathrm{GL}(r)$. We are interested in characterizing the vector space of $G$-equivariant linear maps, which we denote by $\mathrm{Hom}_G(\mathcal{I}, \mathcal{O})$. In the main text we show that the maps $F_{\mathrm{Linear}}^{\boldsymbol{\Phi}_1, \boldsymbol{\Psi}_1, \ldots, \boldsymbol{\Phi}_L, \boldsymbol{\Psi}_L}$, defined by

$$F_{\mathrm{Linear}}^{\boldsymbol{\Phi}_1, \boldsymbol{\Psi}_1, \ldots, \boldsymbol{\Phi}_L, \boldsymbol{\Psi}_L} (U_1, V_1, \ldots, U_L, V_L) = (\boldsymbol{\Phi}_1 U_1, \boldsymbol{\Psi}_1 V_1, \ldots, \boldsymbol{\Phi}_L U_L, \boldsymbol{\Psi}_L V_L) \qquad (10)$$

are all $G$-equivariant. In other words, we showed that

$$\mathcal{L} := \left\{ F_{\mathrm{Linear}}^{\boldsymbol{\Phi}_1, \boldsymbol{\Psi}_1, \ldots, \boldsymbol{\Phi}_L, \boldsymbol{\Psi}_L} \mid \boldsymbol{\Phi}_i \in \mathbb{R}^{n_i' \times n_i}, \boldsymbol{\Psi}_i \in \mathbb{R}^{m_i' \times m_i} \right\} \subseteq \mathrm{Hom}_G(\mathcal{I}, \mathcal{O}). \qquad (11)$$

$\mathcal{L}$ is a linear subspace of dimension $\sum_{i=1}^{L} n_i n_i' + \sum_{i=1}^{L} m_i m_i'$, so in order to prove that $\mathcal{L} = \mathrm{Hom}_G(\mathcal{I}, \mathcal{O})$, it's enough to show that $\dim(\mathrm{Hom}_G(\mathcal{I}, \mathcal{O})) = \sum_{i=1}^{L} n_i n_i' + \sum_{i=1}^{L} m_i m_i'$. Since $\mathcal{I}$ and $\mathcal{O}$ are direct sums of representations, Lemma 1 implies that the dimension of $\mathrm{Hom}_G(\mathcal{I}, \mathcal{O})$ is the sum of the dimensions of the constituents:

$$\dim(\mathrm{Hom}_G(\mathcal{I}, \mathcal{O})) = \sum_{i=1}^{L} \sum_{i'=1}^{L} \Big( \dim\left( \mathrm{Hom}_G(\mathcal{U}_i, \tilde{\mathcal{U}}_{i'}) \right) + \dim\left( \mathrm{Hom}_G(\mathcal{U}_i, \tilde{\mathcal{V}}_{i'}^*) \right) + $$
$$\dim\left( \mathrm{Hom}_G(\mathcal{V}_i^*, \tilde{\mathcal{U}}_{i'}) \right) + \dim\left( \mathrm{Hom}_G(\mathcal{V}_i^*, \tilde{\mathcal{V}}_{i'}^*) \right) \Big). \qquad (12)$$

Next, note that $\mathcal{U}_i$, $\mathcal{V}_i^*$, $\tilde{\mathcal{U}}_i$, and $\tilde{\mathcal{V}}_i^*$ all decompose into irreducible representations

$$\mathcal{U}_i = \bigoplus_{j=1}^{n_i} \mathcal{U}_i^j, \ \mathcal{V}_i^* = \bigoplus_{j=1}^{m_i} (\mathcal{V}_i^j)^*, \ \tilde{\mathcal{U}}_i = \bigoplus_{j=1}^{n_i'} \tilde{\mathcal{U}}_i^j, \ \tilde{\mathcal{V}}_i^* = \bigoplus_{j=1}^{m_i'} (\tilde{\mathcal{V}}_i^j)^*, \qquad (13)$$

each of which is isomorphic to the standard representation of the $i$-th copy of $\mathrm{GL}(r)$ on either $\mathbb{R}^r$ (for $\mathcal{U}_i^j$ and $\tilde{\mathcal{U}}_i^j$) or $(\mathbb{R}^r)^*$ (for $(\mathcal{V}_i^j)^*$ and $(\tilde{\mathcal{V}}_i^j)^*$). We can think of $\mathcal{U}_i^j$ as the space spanned by the $j$-th row in the input $U_i$ matrix and $(\mathcal{V}_i^j)^*$ as the space spanned by the $j$-th column of the input $V_i^\top$ matrix. This decomposition gives us

$$\dim\left( \mathrm{Hom}_G(\mathcal{U}_i, \tilde{\mathcal{U}}_{i'}) \right) = \sum_{j=1}^{n_i} \sum_{j'=1}^{n_i'} \dim\left( \mathrm{Hom}_G(\mathcal{U}_i^j, \tilde{\mathcal{U}}_{i'}^{j'}) \right),$$

$$\dim\left( \mathrm{Hom}_G(\mathcal{U}_i, \tilde{\mathcal{V}}_{i'}^*) \right) = \sum_{j=1}^{n_i} \sum_{j'=1}^{m_i'} \dim\left( \mathrm{Hom}_G(\mathcal{U}_i^j, (\tilde{\mathcal{V}}_{i'}^{j'})^*) \right),$$

$$\dim\left( \mathrm{Hom}_G(\mathcal{V}_i^*, \tilde{\mathcal{U}}_{i'}) \right) = \sum_{j=1}^{m_i} \sum_{j'=1}^{n_i'} \dim\left( \mathrm{Hom}_G((\mathcal{V}_i^j)^*, \tilde{\mathcal{U}}_{i'}^{j'}) \right),$$

$$\dim\left( \mathrm{Hom}_G(\mathcal{V}_i^*, \tilde{\mathcal{V}}_{i'}^*) \right) = \sum_{j=1}^{m_i} \sum_{j'=1}^{m_i'} \dim\left( \mathrm{Hom}_G((\mathcal{V}_i^j)^*, (\tilde{\mathcal{V}}_{i'}^{j'})^*) \right). \qquad (14)$$

Since these are all irreducible representations, Schur's lemma (Fulton & Harris, 1991) implies that the dimension of the space of $G$-equivariant maps is either 1 (if the representations are isomorphic) or 0 (if they are not). Notice that,

- $\mathcal{U}_i^j$ and $\tilde{\mathcal{U}}_{i'}^{j'}$ are isomorphic as $G$-representations if and only if $i = i'$ (if $i \neq i'$ different copies of $\mathrm{GL}(r)$ act on $\mathcal{U}_i^j$ and $\tilde{\mathcal{U}}_{i'}^{j'}$).

- $(\mathcal{V}_i^j)^*$ and $(\tilde{\mathcal{V}}_{i'}^{j'})^*$ are isomorphic as $G$-representations if and only if $i = i'$ (if $i \neq i'$ different copies of $\mathrm{GL}(r)$ act on $(\mathcal{V}_i^j)^*$ and $(\tilde{\mathcal{V}}_{i'}^{j'})^*$).

- $\mathcal{U}_i^j$ is never isomorphic to $(\tilde{\mathcal{V}}_{i'}^{j'})^*$ and $(\mathcal{V}_i^j)^*$ is never isomorphic to $\tilde{\mathcal{U}}_{i'}^{j'}$.

Therefore, together with Equation 12 and Equation 14 we get

$$
\begin{aligned}
\dim\left(\mathrm{Hom}_G(\mathcal{I}, \mathcal{O})\right) &= \sum_{i=1}^{L} \sum_{i'=1}^{L} \left(n_i n_{i'}' \cdot \mathbf{1}_{i=i'} + m_i m_{i'}' \cdot \mathbf{1}_{i=i'}\right) \\
&= \sum_{i=1}^{L} n_i n_i' + \sum_{i=1}^{L} m_i m_i'
\end{aligned}
\tag{15}
$$

concluding the proof. $\qquad\square$

## B  PROOF OF INVARIANCES: THEOREM 1

We prove invariance (or lack thereof) of each model one by one. For this section, consider arbitrary inputs $\mathbf{UV} = (U_1, V_1, \ldots, U_L, V_l)$, where $U_i \in \mathbb{R}^{n_i \times r}$ and $V_i \times \mathbb{R}^{m_i \times r}$. Further, choose invertible $\mathbf{R} = (R_1, \ldots, R_L) \in \mathrm{GL}(r)^L$. Denote the application of $\mathbf{R}$ on $\mathbf{UV}$ by $\mathbf{R} \star \mathbf{UV} = (U_1 R_1, V_1 R_1^{-\top}, \ldots, U_L R_L, V_L R_L^{-\top})$ To show that a function $f$ is GL invariant, we will show that $f(\mathbf{R} \star \mathbf{UV}) = f(\mathbf{UV})$. Note that, since $\mathrm{O}(r) \subset \mathrm{GL}(r)$, if we show that a function is GL-invariant, then it is also O-invariant.

**MLP.**  We will show that the simple MLP is neither O-invariant or GL-invariant. It suffices to show that it is not O-invariant. As a simple example, let $\mathrm{MLP}(U, V) = U_{1,1}$ output the top-left entry of $U$. Then let $U$ be a matrix with 1 in the top-left corner and 0 elsewhere. Further, let $P$ be the permutation matrix that swaps the first and second entries of its input. Then $\mathrm{MLP}(UP, VP) = 0 \neq \mathrm{MLP}(U, V) = 1$. As permutation matrices are orthogonal, $P \in \mathrm{O}(r)$, so the MLP is indeed not $\mathrm{O}(r)$ invariant.

**MLP + O-Align.**  We will show that the O-alignment approach is O-invariant on all but a Lebesgue-measure-zero set. For simplicity, let $L = 1$, $U \in \mathbb{R}^{n \times r}$, and $V \in \mathbb{R}^{m \times r}$. Let $\mathbf{U} \in \mathbb{R}^{n \times r}$ and $\mathbf{V} \in \mathbb{R}^{m \times r}$ be the template matrices, which we assume are full rank (the full rank matrices are a Lebesgue-dense set, so this is an allowed assumption). Recall that we canonicalize $U, V$ as $\rho(U, V) = (UQ, VQ)$, where:

$$
Q = \underset{Q \in O(r)}{\arg\min} \|UQ - \mathbf{U}\|_F^2 + \|VQ - \mathbf{V}\|_F^2.
\tag{16}
$$

We call any solution to this problem a *canonicalizing matrix* for $(U, V)$. This can be equivalently written as

$$
Q = \underset{Q \in O(r)}{\arg\min} \left\| \begin{bmatrix} U \\ V \end{bmatrix} Q - \begin{bmatrix} \mathbf{U} \\ \mathbf{V} \end{bmatrix} \right\|_F^2.
\tag{17}
$$

Let $M = U^\top \mathbf{U} + V^\top \mathbf{V}$. Then a global minimum of this problem is $Q = AB^\top$, where $A\Sigma B^\top$ is an SVD of $M$. If $M$ has distinct singular values, then $A$ and $B$ are unique up to sign flips, and $AB^\top$ is in fact unique. Thus, if $M$ has unique singular values, $(UQ, VQ)$ is unique. We assume from here that $M$ has distinct singular values, as the set of all $M$ that do form a Lebesgue-dense subset of $\mathbb{R}^{r \times r}$ (and thus this is satisfies for Lebesgue-almost-every $U$ and $V$).

Now, to show O-invariance, let $\tilde{Q} \in \mathrm{O}(r)$. We will show that $\rho(U\tilde{Q}, V\tilde{Q}) = \rho(U, V)$.

$$\operatorname*{arg\,min}_{Q \in O(r)} \left\| \begin{bmatrix} U\tilde{Q} \\ V\tilde{Q} \end{bmatrix} Q - \begin{bmatrix} \mathbf{U} \\ \mathbf{V} \end{bmatrix} \right\|_F^2 = \operatorname*{arg\,min}_{Q' \in O(r)} \left\| \begin{bmatrix} U \\ V \end{bmatrix} Q' - \begin{bmatrix} \mathbf{U} \\ \mathbf{V} \end{bmatrix} \right\|_F^2 \tag{18}$$

because the orthogonal matrices are closed under multiplication. Thus, if $Q$ is a canonicalizing matrix for $(U, V)$, then $\tilde{Q}^\top Q$ is a canonicalizing matrix for $(U\tilde{Q}, V\tilde{Q})$. Moreover, we have that $\tilde{Q}^\top U^\top \mathbf{U} + \tilde{Q}^\top V^\top \mathbf{V} = \tilde{Q}^\top M$, so this has the same singular values as $M$ (as orthogonal matrices don't affect singular values); this means that the singular values are distinct, so there is a unique canonicalizing matrix for $(U\tilde{Q}, V\tilde{Q})$. This means that the canonicalization matrix must be equal to $\tilde{Q}^\top Q$, so that

$$\rho(U\tilde{Q}, V\tilde{Q}) = (U\tilde{Q}\tilde{Q}^\top Q, V\tilde{Q}\tilde{Q}^\top Q) = (UQ, VQ) = \rho(U, V). \tag{19}$$

As this argument holds for Lebesgue-almost-every $U$ and $V$, we have shown O-invariance of MLP + O-Align.

Finally, we have to show that this LoL model is not GL-invariant. To do this, let the MLP approximate the Frobenius norm function on its first input, so $\mathrm{MLP}(U, V) \approx \|U\|$. Consider any $U$ that is nonzero, and let $a > 2$ be a scalar. Then $aI \in \mathrm{GL}(r)$. Also, we have that $\rho(U, V) = (UQ, VQ)$ for some $Q \in \mathrm{O}(r)$, so this canonicalization does not affect the Frobenius norm of the first entry. Thus, we have that

$$\mathrm{MLP}(\rho(U, V)) \approx \|U\| \neq a\|U\| \approx \mathrm{MLP}(\rho(aU, (1/a)V)). \tag{20}$$

So MLP + O-Align is not invariant under GL.

**MLP + SVD.** We show this is GL-invariant. The output of this model $f$ on $\mathbf{UV}$ can be written as

$$\mathrm{MLP}(\sigma_1(U_1 V_1^\top), \ldots, \sigma_r(U_1 V_1^\top), \ldots, \sigma_1(U_L V_L^\top), \ldots, \sigma_r(U_L V_L^\top)). \tag{21}$$

Where $\sigma_j(U_i V_i^\top)$ is the $j$th singular value of $U_i V_i^\top$. Thus, we can compute that:

$$f(\mathbf{R} \star \mathbf{UV}) = \mathrm{MLP}(\sigma_1(U_1 R_1 R_1^{-1} V_1^\top), \ldots, \sigma_r(U_L R_L R_L^{-1} V_L^\top)) \tag{22}$$
$$= \mathrm{MLP}(\sigma_1(U_1 V_1^\top), \ldots, \sigma_r(U_L V_L^\top)) \tag{23}$$
$$= f(\mathbf{UV}). \tag{24}$$

**MLP + Dense.** We show this is GL-invariant. We can simply see that

$$\mathrm{MLP}(\mathbf{R} \star \mathbf{UV}) = \mathrm{MLP}(U_1 R_1 R_1^{-1} V_1^\top, \ldots, U_L R_L R_L^{-1} V_L^\top) \tag{25}$$
$$= \mathrm{MLP}(U_1 V_1^\top, \ldots, U_L V_L^\top) \tag{26}$$
$$= \mathrm{MLP}(\mathbf{UV}). \tag{27}$$

**GL-net.** We show this is GL-invariant. First, assume that the equivariant linear layers are GL-equivariant, and the equivariant nonlinearities are GL-equivariant. Note that the invariant head of GL-net is a special case of MLP + Dense, which we have already proven to be invariant. Further, an invariant function composed with an equivariant function is invariant, so we are done.

We only need to prove that the nonlinearities are GL-equivariant, because we have already proven that the equivariant linear layers are GL-equivariant in the main text. Let $\sigma : \mathbb{R} \to \mathbb{R}$ be any real function, and recall that the GL-equivariant nonlinearity takes the following form on $U \in \mathbb{R}^{n \times r}$, $V \in \mathbb{R}^{m \times r}$:

$$\sigma_{\mathrm{GL}}(U, V) = (\tilde{U}, \tilde{V}), \qquad \tilde{U}_i = \sigma\Big(\sum_j (UV^\top)_{ij}\Big) U_i, \qquad \tilde{V}_i = \sigma\Big(\sum_j (UV^\top)_{ji}\Big) V_i, \tag{28}$$

where $\tilde{U} \in \mathbb{R}^{n \times r}$, $\tilde{V} \in \mathbb{R}^{m \times r}$, and for example $U_i \in \mathbb{R}^r$ denotes the $i$th row of $U_i$.

**Lemma 2.** *For any real function $\sigma : \mathbb{R} \to \mathbb{R}$, the function $\sigma_{\mathrm{GL}}$ defined in equation 28 is GL equivariant.*

*Proof.* Let $U \in \mathbb{R}^{n \times r}, V \in \mathbb{R}^{m \times r}$, and $R \in \mathrm{GL}(r)$ be arbitrary. Denote the output of the nonlinearity on the transformed weights as $\sigma_{\mathrm{GL}}(UR, VR^{-\top}) = (\tilde{U}^{(R)}, \tilde{V}^{(R)})$. Then we have that

$$\tilde{U}_i^{(R)} = \sigma\Big(\sum_j (URR^{-1}V^\top)_{ij}\Big) R^\top U_i = R^\top \left[\sigma\Big(\sum_j (UV^\top)_{ij}\Big) U_i\right] = R^\top U_i \qquad (29)$$

$$\tilde{V}_i^{(R)} = \sigma\Big(\sum_j (URR^{-1}V^\top)_{ji}\Big) R^{-1} V_i = R^{-1} \left[\sigma\Big(\sum_j (UV^\top)_{ji}\Big) V_i\right] = R^{-1} V_i. \qquad (30)$$

In matrix form, this means that $\tilde{U}_i^{(R)} = \tilde{U}_i R$ and $\tilde{V}_i^{(R)} = \tilde{V}_i R^{-\top}$. In other words,

$$\sigma_{\mathrm{GL}}(UR, VR^{-1}) = R \star \sigma_{\mathrm{GL}}(U, V), \qquad (31)$$

which is the definition of GL-equivariance, so we are done. $\qquad\square$

## C    PROOF OF EXPRESSIVITY: THEOREM 2

In this section we formally restate and prove Theorem 2. We start by defining full rank GL universality for LoL models.

**Definition C.1** (Full rank GL-universality). *Let $\mathcal{D} = \{(U_1, V_1, \ldots, U_L, V_L) \mid U_i \in \mathbb{R}^{n_i \times r}, V_i \in \mathbb{R}^{m_i}, \mathrm{rank}(U_i) = \mathrm{rank}(V_i) = r\}$ be the set of LoRA updates of full rank. A LoL architecture is called full rank GL-universal if for every GL-invariant function $f : \mathcal{D} \to \mathbb{R}$, every $\epsilon > 0$, and every compact set $K \subset \mathcal{D}$, there is a model $f^{\mathrm{LoL}}$ of said architecture that approximates $f$ on $K$ up to $\epsilon$:*

$$\sup_{\boldsymbol{X} \in K} |f^{\mathrm{LoL}}(\boldsymbol{X}) - f(\boldsymbol{X})| < \epsilon. \qquad (32)$$

Note that the set $\mathcal{D}$ of full-rank LoRA updates is Lebesgue-dense (its compliment has measure 0).

**Theorem 3** (Formal restatement of Theorem 2). *The MLP, MLP + O-Align, MLP + Dense, and GL-net LoL architectures are all full rank GL universal.*

The universality of MLP and MLP + O-Align models follows from the universal approximation thereom for MLPs (Hornik et al., 1989). To prove Theorem 3 for MLP + Dense and GL-net we use the following result from Dym & Gortler (2024).

**Proposition 2** (Proposition 1.3 from Dym & Gortler (2024)). *Let $\mathcal{M}$ be a topological space, and $G$ a group which acts on $\mathcal{M}$. Let $K \subset \mathcal{M}$ be a compact set, and let $f^{\mathrm{inv}} : \mathcal{M} \to \mathbb{R}^N$ be a continuous $G$-invariant map that separates orbits. Then for every continuous invariant function $f : \mathcal{M} \to \mathbb{R}$ there exists some continuous $f^{\mathrm{general}} : \mathbb{R}^N \to \mathbb{R}$ such that $f(x) = f^{\mathrm{general}}(f^{\mathrm{inv}}(x)), \forall x \in K$.*

In order to use the proposition above for MLP + Dense LoL models, we need to show that multiplying out the LoRA updates separates GL orbits.

**Lemma 3.** *The function*

$$f^{\mathrm{mul}}(U_1, V_1, \ldots, U_L, V_L) = (U_1 V_1^\top, \ldots, U_L V_L^\top),$$

*separates GL-orbits in $\mathcal{D}$. That is, if $(U_1, V_1, \ldots, U_L, V_L)$ and $(U_1', V_1', \ldots, U_L', V_L')$ are in different $G$-orbits then $f^{\mathrm{mul}}(U_1, V_1, \ldots, U_L, V_L) \neq f^{\mathrm{mul}}(U_1', V_1', \ldots, U_L', V_L')$.*

*Proof.* We prove the contrapositive. Let $(U_1, V_1, \ldots, U_L, V_L)$ and $(U_1', V_1', \ldots, U_L', V_L')$ be LoRA updates such that $f^{\mathrm{mul}}(U_1, V_1, \ldots, U_L, V_L) = f^{\mathrm{mul}}(U_1', V_1', \ldots, U_L', V_L')$, i.e. $\forall i \in \{1, \ldots, L\}$

$$U_i V_i^\top = U_i' V_i'^\top. \qquad (33)$$

Since $U_i$, $V_i$, $U_i'$, and $V_i'$ are of rank $r$, their corresponding Gram matrices $U_i^\top U_i$, $V_i^\top V_i$, $U_i'^\top U_i', V_i'^\top V_i' \in \mathbb{R}^{r \times r}$, are also of rank $r$ and are thus invertible. Multiplying both sides of Equation 33 by $V_i(V_i^\top V_i)^{-1}$ from the right, we get

$$U_i \underbrace{V_i^\top V_i (V_i^\top V_i)^{-1}}_{} = U_i' \underbrace{V_i'^\top V_i (V_i^\top V_i)^{-1}}_{R_i}. \qquad (34)$$

Substituting $U_i' R_i$ back to Equation 33 we get

$$U_i' R_i V_i^\top = U_i' V_i'^\top.$$

Multiplying by $(U_i'^\top U_i')^{-1} U_i'^\top$ from the left gives

$$\underline{(U_i'^\top U_i')^{-1} U_i'^\top U_i'} R_i V_i^\top = \underline{(U_i'^\top U_i')^{-1} U_i'^\top U_i'} V_i'^\top.$$

Therefore, to prove $(U_1, V_1, \ldots, U_L, V_L)$ and $(U_1', V_1', \ldots, U_L', V_L')$ are in the same orbit all we need to do is show that $R_i \in \mathrm{GL}(r)$. To do so it's enough to show that $V_i'^\top V_i$ is invertible. And indeed, starting from Equation 33

$$U_i V_i^\top = U_i' V_i'^\top$$

*(Multiply both sides by $(U_i^\top U_i)^{-1} U_i^\top$ from the left)*

$$(U_i^\top U_i)^{-1} U_i^\top U_i V_i^\top = (U_i^\top U_i)^{-1} U_i^\top U_i' V_i'^\top$$

*(Simplify left side: $(U_i^\top U_i)^{-1} U_i^\top U_i = I$)*

$$V_i^\top = (U_i^\top U_i)^{-1} U_i^\top U_i' V_i'^\top \tag{35}$$

*(Multiply both sides by $V_i$ from the right)*

$$V_i^\top V_i = (U_i^\top U_i)^{-1} U_i^\top U_i' V_i'^\top V_i$$

*(Multiply both sides by $(V_i^\top V_i)^{-1}$ from the left)*

$$I = (V_i^\top V_i)^{-1} (U_i^\top U_i)^{-1} U_i^\top U_i' V_i'^\top V_i.$$

Therefore, $R_1, \ldots, R_L \in \mathrm{GL}(r)$, $U_i = U_i' R$, and $V_i = V_i' R_i^{-\top}$, implying that $(U_1, V_1, \ldots, U_L, V_L)$ and $(U_1', V_1', \ldots, U_L', V_L')$ are in the same $G$-orbit. $\qquad\square$

We are now ready to prove Theorem 3.

*Proof of theorem 3.* Let $f : \mathcal{D} \to \mathbb{R}$ be a continuous GL-invariant function, let $K \subset \mathcal{D}$ be a compact set and fix $\epsilon > 0$.

1. **MLP.** Since $K$ is compact and $f$ is continuous, universality follows from the universal approximation theorem for MLPs (Hornik et al., 1989).

2. **MLP + O-Align.** Let $f^{\mathrm{align}}$ be the O-canonicalization function. $f^{\mathrm{align}}$ is continuous so $f^{\mathrm{align}}(K)$ is compact. and let $f^{\mathrm{MLP}}$ be an MLP that approximates $f$ up-to $\epsilon$ on $f^{\mathrm{align}}(K)$.

$$\sup_{\boldsymbol{X} \in K} |f^{\mathrm{MLP}}(f^{\mathrm{align}}(\boldsymbol{X})) - f(\boldsymbol{X})| = \sup_{\boldsymbol{X} \in K} |f^{\mathrm{MLP}}(f^{\mathrm{align}}(\boldsymbol{X})) - f(f^{\mathrm{align}}(\boldsymbol{X}))|$$

$$= \sup_{\boldsymbol{Y} \in f^{\mathrm{align}}(K)} |f^{\mathrm{MLP}}(\boldsymbol{Y}) - f(\boldsymbol{Y})| < \epsilon.$$

The first equality holds since $f$ is GL-invariant, and in particular O-invariant.

3. **MLP + Dense.** From Lemma 3 we know that $f^{\mathrm{mul}}$ separates orbits. It's additionally clear that $f^{\mathrm{mul}}$ is continuous and $G$-invariant. Therefore, using Proposition 2 there exists a function $f^{\mathrm{general}} : \mathbb{R}^N \to \mathbb{R}$ such that $f \equiv f^{\mathrm{general}} \circ f^{\mathrm{mul}}$ on $K$. Since $f^{\mathrm{mul}}$ is continuous, $f^{\mathrm{mul}}(K)$ is also compact and we can use the universal approximation theorem of MLPs for $f^{\mathrm{genral}}$ on $f^{\mathrm{mul}}(K)$. Therefore, there exists an MLP $f^{\mathrm{MLP}}$ such that

$$\sup_{\boldsymbol{X} \in K} |f^{\mathrm{MLP}}(f^{\mathrm{mul}}(\boldsymbol{X})) - f(\boldsymbol{X})| = \sup_{\boldsymbol{X} \in K} |f^{\mathrm{MLP}}(f^{\mathrm{mul}}(\boldsymbol{X})) - f^{\mathrm{general}}(f^{\mathrm{mul}}(\boldsymbol{X}))|$$

$$= \sup_{\boldsymbol{Y} \in f^{\mathrm{mul}}(K)} |f^{\mathrm{MLP}}(\boldsymbol{Y}) - f^{\mathrm{general}}(\boldsymbol{Y})| < \epsilon \tag{36}$$

4. **GL-net.** Since we proved MLP + Dense is universal, it's enough to show that GL-net can implement MLP + Dense. If we take a GL-net with no equivariant layers (or equivalently a single equivariant layer that implements the identity by setting $\boldsymbol{\Phi}_i = I_{n_i}$, $\boldsymbol{\Psi}_i = I_{m_i}$) and apply the invariant head directly to the input, the resulting model is exactly MLP + Dense.

$\qquad\square$

# D EXPERIMENTAL DETAILS

## D.1 DIFFUSION MODEL LoRA DATASETS

### D.1.1 CELEBA FINETUNING

Table 6: Hyperparameter distributions for the 3,900 different LoRA diffusion model finetunes we trained. $U(S)$ denotes the uniform distribution over a set $S$.

| Hyperparameter | Distribution |
|---|---|
| Learning rate | $U(\{10^{-4}, 3 \cdot 10^{-4}, 10^{-3}, 3 \cdot 10^{-3}\})$ |
| Train Steps | $U(\{100, 133, 167, 200\})$ |
| Batch Size | $U(\{1, 2\})$ |
| Prompt | $U(\{\text{"Celebrity", "Person", "thing", "skd"}\})$ |
| Rank | 4 |

We finetune 3,900 models on various celebrities in the CelebA dataset (Liu et al., 2015) using the DreamBooth personalization method (Ruiz et al., 2023). Each LoRA is personalized to one celebrity by finetuning on 21 images of that celebrity. Every LoRA is trained starting from a different random initialization, with hyperparameters randomly sampled from reasonable distributions — see Table 6 for the distributions.

**CLIP score prediction.** For CLIP score prediction, we use the following prompts, the first thirty of which are from PartiPrompts Yu et al. (2022), and the last three of which are written to include words relevant to prompts the models are finetuned on. We use a fixed random seed of 42 for image generation. For predicting CLIP scores, GL-net takes the mean of each matrix product $U_i V_i^\top$ in the invariant head. This is equivalent to using an equivariant hidden dimension of 1.

1. 'a red sphere on top of a yellow box',

2. 'a chimpanzee sitting on a wooden bench',

3. 'a clock tower',

4. 'toy cars',

5. 'a white rabbit in blue jogging clothes doubled over in pain while a turtle wearing a red tank top dashes confidently through the finish line',

6. 'a train going to the moon',

7. 'Four cats surrounding a dog',

8. 'the Eiffel Tower in a desert',

9. 'The Millennium Wheel next to the Statue of Liberty. The Sagrada Familia church is also visible.',

10. 'A punk rock squirrel in a studded leather jacket shouting into a microphone while standing on a stump and holding a beer on dark stage.',

11. 'The Statue of Liberty surrounded by helicopters',

12. 'A television made of water that displays an image of a cityscape at night.',

13. 'a family on a road trip',

14. 'the mona lisa wearing a cowboy hat and screaming a punk song into a microphone',

15. 'a family of four posing at Mount Rushmore',

16. 'force',

17. 'an oil surrealist painting of a dreamworld on a seashore where clocks and watches appear to be inexplicably limp and melting in the desolate landscape. a table on the left, with a golden watch swarmed by ants. a strange fleshy creature in the center of the painting',

18. 'Downtown Austin at sunrise. detailed ink wash.',

19. 'A helicopter flies over Yosemite.',

20. 'A giraffe walking through a green grass covered field',
21. 'a corgi's head',
22. 'portrait of a well-dressed raccoon, oil painting in the style of Rembrandt',
23. 'a volcano',
24. 'happiness',
25. "the words 'KEEP OFF THE GRASS' on a black sticker",
26. 'A heart made of wood',
27. 'a pixel art corgi pizza',
28. 'two wine bottles',
29. 'A funny Rube Goldberg machine made out of metal',
30. 'a horned owl with a graduation cap and diploma',
31. 'A celebrity in a park',
32. 'A person on the beach',
33. 'A thing in a city'

**CelebA Attribute Prediction.** As mentioned in Section 5.2.2, we only predict the five least noisy CelebA attributes, as measured by Lingenfelter et al. (2022). Recall that each LoRA in CelebA-LoRA is trained on 21 images of a given celebrity. The attribute labels can vary across these 21 images, so we say the ground truth attribute label of the LoRA is the majority label across these 21 images.

### D.1.2 IMAGENETTE FINETUNING

For Imagenette (Howard, 2019), we finetune 2,046 models. In particular, for each nonempty subset of the 10 Imagenette classes, we finetune 2 models using 1 image from each present class. We use a rank of 32, in part so that we can test how well LoL models perform on larger ranks than the other experiments.

Table 7: Hyperparameters for the dataset of 2,046 different LoRA diffusion model finetunes we trained on Imagenette (Imagenette-LoRA).

| Hyperparameter | Value |
|---|---|
| Learning rate | $3 \cdot 10^{-4}$ |
| Train Steps | 150 |
| Batch Size | 1 |
| Prompt | $sks\_photo$ |
| Rank | 32 |

### D.2 LANGUAGE MODEL LoRA DATASET

Here, we describe the details of our Qwen2-ARC-LoRA dataset of trained language model LoRAs.

For training data, we use the ARC training set (Clark et al., 2018), using both the easy and challenge splits. First, we hold out a fixed validation set sampled from this set to compute the validation loss on. Each training data point originates from one of 20 sources (e.g. some questions come from Ohio Achievement Tests, and some come from the Virginia Standards of Learning). For each LoRA finetuning run, we sample a random subset of these sources to use as training data (except we do not ever omit the Mercury source, which contains many more data points than the other sources). To do this, we sample a random integer in $s \in [1, 19]$, then choose a random size-$s$ subset of the 19 possibly-filtered sources to drop.

For each LoRA finetuning run, we sample random hyperparameters: learning rate, weight decay, number of epochs, batch size, LoRA dropout, and the data sources to filter. The distributions from which we sample are shown in Table 8, and were chosen to give reasonable but varied performance across different runs. All trained LoRAs were of rank 4, and we only applied LoRA to tune the key and value projection matrices of each language model.

Table 8: Hyperparameter distributions for the 2,000 different LoRA language model finetunes we trained (Qwen2-ARC-LoRA). $U(S)$ denotes the uniform distribution over a set $S$.

| Hyperparameter | Distribution |
|---|---|
| Learning rate | $10^{U([-5,-3])}$ |
| Weight decay | $10^{U([-6,-2])}$ |
| Epochs | $U(\{2, 3, 4\})$ |
| Batch Size | $U(\{32, 64, 128\})$ |
| LoRA Dropout | $U([0, .1])$ |
| Filtered sources | $U(\text{sources})$ |

### D.3 DATASET SIZE PREDICTION

Here we describe theoretical and experimental details relating to the implicit strategies that GL-net and MLP + SVD may employ in dataset size prediction as in Table 4.

Suppose that a model $f_{\text{class}}$ trained to predict Imagenette class presence of LoRAs (as in Table 3) has test accuracy $p$, while each of the 10 classes is present with probability .5. For LoRA weights $x$, $f_{\text{class}}(x)_i = 1$ if the model predicts that class $i$ is in the finetuning dataset of $x$. Define the corresponding dataset-size prediction model $f_{\text{size}}(x) = \sum_{i=1}^{10} f_{\text{class}}(x)_i$ That is, $f_{\text{size}}$ predicts whether each class is in the dataset, sums up these predictions, and then outputs the sum as the expected dataset size.

Assuming $f_{\text{class}}$ is equally likely to provide false-negative or false-positive predictions for each class, each of its 10 outputs has probability $1 - p$ of being incorrect. Consider an input LoRA $x$ with a dataset size $s$. Then $f_{\text{size}}(x) = \hat{y}_1 + \hat{y}_2 + \ldots \hat{y}_{10}$, where $\hat{y}_i$ is equal to $y_i$ with probability $p$, and $1 - y_i$ with probability $1 - p$. So,

$$f_{\text{size}}(x) = s \iff |\{i \mid y_i = 1 \wedge \hat{y}_i = 0\}| = |\{i \mid y_i = 0 \wedge \hat{y}_i = 1\}| \qquad (37)$$

The cardinality of the left set is distributed as $\text{binom}(s, 1-p)$, while the cardinality of the right set is distributed as $\text{binom}(10-s, 1-p)$. So, $\mathbb{P}(f_{\text{size}}(x) = s|s) = \mathbb{P}(\beta_1 = \beta_2)$, for $\beta_1 \sim \text{binom}(s, 1-p)$ and $\beta_2 \sim \text{binom}(10 - s, 1 - p)$. Thus,

$$\mathbb{P}_{(x,s)\sim\text{data}}(f_{\text{size}}(x) = s) = \sum_{\eta=0}^{10} [\mathbb{P}(\text{binom}(\eta, 1-p) = \text{binom}(10-\eta, 1-p) \mid \eta) \cdot \mathbb{P}(\eta)] \qquad (38)$$

Table 3 shows that for GL-net, $f_{\text{class}}$ is correct with probability $p = .904$. So, we have $1 - p = .096$.

Using a Python script to evaluate equation 38, we find that the probability that $f_{\text{size}}(x)$ is correct is 46.1%, which almost exactly matches the observed probability of 46.8% that we obtain from training GL-net to predict the dataset size in Table 4.

We further test our hypothesis empirically by training GL-net on Imagenette class prediction (as in Table 3) and then summing up its class predictions to predict dataset size. On the Imagenette size-predictoin test set, GL-net with this sum strategy achieves an accuracy of $43.5 \pm 2.1\%$. This means GL-net is likely learning a function with underlying mechanism similar to $f_{\text{size}}$, where it predicts the presence of each class and then sums those predictions to output the total dataset size.

On the other hand, MLP + SVD correctly predicts class presence with probability $p = .656$. Predicting classes and summing up its individual predictions would give MLP + SVD a theoretical accuracy of 21% by equation 38, which is significantly worse than its true performance of 73.1%. This suggests that MLP + SVD likely looks a characteristics other than class presence to determine dataset size.

## E LORA VARIANTS AND THEIR SYMMETRIES

In this section, we describe various LoRA variants that have been proposed for parameter-efficient finetuning. We also discuss their symmetries, and how one may process these LoRA variants with LoL models.

**Training Modifications.** Several LoRA variants such as PiSSA (Meng et al., 2024) and LoRA+ (Hayou et al., 2024) have the same type of weight decomposition as the original LoRA, but with different initialization or training algorithms. These variants have the same exact symmetries as standard LoRA, so they can be processed in exactly the same way with our LoL models.

**DoRA (Weight-Decomposed Low-Rank Adaptation).** Liu et al. (2024) decompose the parameter updates into magnitude and directional components. For base weights $W \in \mathbb{R}^{n \times m}$, DoRA finetuning learns the standard low rank weights $U \in \mathbb{R}^{n \times r}$ and $V \in \mathbb{R}^{m \times r}$ along with a magnitude vector $\mathbf{m} \in \mathbb{R}^m$ so that the new finetuned weights are given by

$$(W + UV^\top)\mathrm{Diag}\left(\frac{\mathbf{m}}{\|W + UV^\top\|_c}\right), \tag{39}$$

where $\|\cdot\|_c : \mathbb{R}^{n \times m} \to \mathbb{R}^m$ is the norm of each column of the input matrix, the division $\frac{\mathbf{m}}{\|W + UV^\top\|_c}$ is taken elementwise, and $\mathrm{Diag}$ takes vectors in $\mathbb{R}^m$ to diagonal matrices in $\mathbb{R}^{m \times m}$. The vector $\mathbf{m}$ represents the norm of each column of the finetuned matrix. DoRA weights $(U, V, \mathbf{m})$ also have the same invertible matrix symmetry as standard LoRA, where $(UR, VR^{-\top}, \mathbf{m})$ is functionally equivalent to $(U, V, \mathbf{m})$. The $\mathbf{m}$ vector cannot in general be changed without affecting the function. Thus, an LoL model could take as input $\mathbf{m}$ as an invariant feature, for instance by concatenating it to features in an invariant head of GL-net, or concatenating it to the input of an MLP in the other LoL models.

**KronA (Kronecker Adapter).** Edalati et al. (2022) use a Kronecker product structure to decompose the learned weight matrix in a parameter-efficient way. For a weight matrix $W$ of shape $n \times m$, LoKr learns $U \in \mathbb{R}^{n' \times m'}, V \in \mathbb{R}^{n'' \times m''}$ such that the finetuned weight is given by $W + U \otimes V$. Here, we no longer have a large general linear symmetry group. Instead, there is a scale symmetry, where $(U, V)$ is equivalent to $(sU, \frac{1}{s}V)$ for a scalar $s \in \mathbb{R} \setminus \{0\}$, since $(sU) \otimes (\frac{1}{s}V) = U \otimes V$. For LoL tasks, we can use a scale-invariant architecture, as for instance explored by Kalogeropoulos et al. (2024). We can also use GL-net on the flattened $\mathrm{vec}(U) \in \mathbb{R}^{n'm' \times 1}$ and $\mathrm{vec}(V) \in \mathbb{R}^{n''m'' \times 1}$, since $\mathbb{R} \setminus \{0\} = \mathrm{GL}(1)$ is the general linear symmetry group in the special case of rank $r = 1$.

