# OpenReview forum: "Learning on LoRAs: GL-Equivariant Processing of Low-Rank Weight Spaces for Large Finetuned Models"
_ICLR.cc/2025/Conference — Submitted to ICLR 2025_

### Official Review · Reviewer_EBmP · 2024-10-16

**Soundness:** 2
**Presentation:** 2
**Contribution:** 1
**Rating:** 3
**Confidence:** 3

**Summary:**

The work introduces the Learning on LoRAs (LoL) paradigm; LoL is an ML model which has LoRA adapters as input and it outputs predictions about the weights.

**Strengths:**

Shows that LoRA adapters have information in them that can be learned.
I guess the idea of learning a model for LoRA adapters is technically novel, then again the fact that weights encode information and that information can be extracted isn't a wild thought.

**Weaknesses:**

I don't really understand why this is at all useful...

**Questions:**

- The model allows you to predict the accuracy of the model if a set of LoRA weights is applied; how is this useful? Why not just apply the LoRA weights and actually measure the accuracy?

- Why is it useful to predict how many data points the model was fine tuned on... If I have a set of LoRA adapters, I'll try every single one and the set of LoRA weights that does best is the one I'll use (regardless of how many data points it was trained on...).

---

> ### Author Response · Authors · 2024-11-18
>
> We thank the reviewer for taking their time to review our paper. Here we address comments by the reviewer, please let us know if you have any further questions:
>
> > I don't really understand why this is at all useful…
>
> We agree that we could have been clearer in our explanation of the usefulness of LoL models. See our general comment for why we believe LoL models are useful. We specifically address your two specific usefulness-related questions below:
>
> > The model allows you to predict the accuracy of the model if a set of LoRA weights is applied; how is this useful? Why not just apply the LoRA weights and actually measure the accuracy?
>
> As we write in our paper (Section 5.3) “This could be useful in evaluations of finetuned models, as standard evaluations of language models can require a lot of time and resources, whereas our LoL models can evaluate these models with one quick forward pass.“
>
> Quantitatively, for the experiments in the paper, computing CLIP score with our LoL models are up to 53,700 times faster than with standard evaluation, and evaluating an LLM on ARC-C can be 730,000 times faster. More specifically:
> * Evaluating CLIP score of one diffusion model takes 36 ± 2 seconds (660 forward passes, on 33 fixed prompts) when doing a standard evaluation. But it only takes .0133 of a second with one GL-Net forward pass (and .00067 seconds per model at batch size 250).
> * Evaluating ARC-C test accuracy of a Qwen2 1.5B finetune requires 220 seconds (over 1,100 data points) with standard evaluation. But it only takes .0075 of a second with our GL-Net at batch size 1 (and .00003 seconds per model at batch size 250).
>
> Moreover, this trade-off can be even more in the favor of LoL models in different scenarios. For instance, some researchers use thousands of times more prompts to compute CLIP score (Saharia et al. 2022), or 10x larger multiple-choice datasets like MMLU to measure LLM performance. Instead of taking over 30 minutes to evaluate models on these benchmarks, we can use LoL models to evaluate them in a fraction of a second.
>
> We will add these quantitative measurements to our paper.
>
> (Saharia et al. 2022) "Photorealistic text-to-image diffusion models with deep language understanding." NeurIPS 35 (2022). https://arxiv.org/abs/2205.11487.
>
> > Why is it useful to predict how many data points the model was fine tuned on... If I have a set of LoRA adapters, I'll try every single one and the set of LoRA weights that does best is the one I'll use (regardless of how many data points it was trained on...).
>
> As we cite in our paper, the task of predicting dataset size from LoRA weights was considered in prior work https://arxiv.org/abs/2406.19395. They dedicate their whole paper to this task, and they write extensive motivation for the task. In our paper, we also already described the utility of this task (Section 5.2.2): “The success of this task implies a privacy leak, since model developers may sometimes wish to keep the size of their finetuning dataset private. Moreover, the dataset size is a useful quantity to know for data membership inference attacks and model inversion attacks, so accurately predicting dataset size could improve effectiveness of these attacks too“. Basically, model trainers sometimes want to keep properties of their finetuning datasets private; this task explores the extent to which data properties can be inferred from released LoRA weights.

---

> > ### Author Response · Authors · 2024-11-22
> >
> > Hello reviewer EBmP,
> >
> > We are checking back in to ask if you have received our response to your questions. As the response period ends on November 26, we would appreciate if you could ask any further questions now, so that we have time to answer them. Please let us know if you have any more comments, and thank you again for reviewing our paper.

---

### Official Review · Reviewer_snsJ · 2024-10-22

**Soundness:** 2
**Presentation:** 3
**Contribution:** 2
**Rating:** 5
**Confidence:** 4

**Summary:**

In this paper, the authors explore the potential of Learning on LoRAs (LoL), a paradigm where LoRA weights are used as inputs to machine learning models. They fine-tune thousands of text-to-image diffusion models and language models to collect datasets of LoRAs and develop several symmetry-aware, invariant or equivariant LoL models using techniques such as canonicalization, invariant featurization, and equivariant layers.

**Strengths:**

1. Several GL(r) equivariant and invariant neural architectures are proposed to effectively process weights of LoRAs.
2. Creating novel datasets for Learning on LoRAs
3. Authors conduct experiments across various finetuned models.

**Weaknesses:**

1. The detailed information regarding the models, such as the number of parameters and the training costs associated with different model structures and methods, is not sufficiently clear.
2. The motivation behind the task selection for LoL models is not clearly articulated. For instance, the LoL models aim to predict hyperparameters, CLIP scores, and training images of fine-tuned diffusion models using their LoRA weights. However, it is not immediately clear why these tasks are important or how they contribute to advancing the field.
3. Other methods used for comparison in this work are not clearly defined and lacks sufficient discussion.

**Questions:**

1. The LoRA model datasets used in this work are derived from Stable Diffusion V1.4 and Qwen2. A key question is whether the LoL model will remain effective if LoRA models are obtained from different versions, such as Stable Diffusion V2.1 or Qwen V2.5.

2. Is there a clear rationale behind the choice of model architecture for the LoL models? Specifically, is using an MLP sufficient for the tasks at hand, or could other architectures be more effective?

3. The paper mentions that a large number of LoRA models were collected. How many GPU hours during the collection of these LoRA models?

4. The practical usage of the LoL models is not explicitly addressed. Are there any potential real-world applications except the dataset size prediction?

---

> ### Author Response · Authors · 2024-11-18
>
> We thank the reviewer for appreciating our new LoL architectures, novel datasets, and experiments across different types of finetuned models. Here we address some comments of the review:
>
> > 1. … the number of parameters and the training costs associated with different model structures and methods, is not sufficiently clear. … How many GPU hours during the collection of these LoRA models?
>
> Good point. We will add the number of parameters for the LoL models of the experiments in the Appendix. Here is the table for your reference:
>
> | Dataset | Model | Num params |
> | --- | --- | --- |
> | Imagenette-LoRA | MLP | 102m |
> | Imagenette-LoRA | MLP + O-Align | 102m |
> | Imagenette-LoRA | MLP + SVD | 500k |
> | Imagenette-LoRA | MLP + Dense | 748m |
> | Imagenette-LoRA | GL-Net | 40m |
> | Qwen2-ARC-LoRA | MLP  |  102,827,777  |
> | Qwen2-ARC-LoRA | MLP + O-Align  | 102,827,777 |
> | Qwen2-ARC-LoRA | MLP + SVD  | 124,673 |
> | Qwen2-ARC-LoRA | MLP + Dense  | 352,321,905 |
> | Qwen2-ARC-LoRA | GL-Net | 5,340,435 |
> | Llama3.2-ARC-LoRA | MLP  | 234,952,979  |
> | Llama3.2-ARC-LoRA | MLP + O-Align  | 234,952,979  |
> | Llama3.2-ARC-LoRA | MLP + SVD  | 129,299 |
> | Llama3.2-ARC-LoRA | GL-Net | 7,404,819 |
>
> Also, we add GPU-hour estimates for training all of the LoRAs.
>
> | Dataset | Approximate GPU hours |
> | --- | --- |
> | CelebA-LoRA            | ~48 hrs, NVIDIA 3090 chip |
> | CelebA-LoRA (Compute CLIP scores)  | ~48 hrs, NVIDIA 3090 chip |
> | Imagenette-LoRA      | ~48 hrs, NVIDIA 3090 chip |
> | Qwen2-ARC-LoRA   | ~384 hrs, NVIDIA 2080ti chip |
> | Llama3.2-ARC-LoRA (new)   | ~448 hrs, NVIDIA 2080ti chip |
>
> > 2. The motivation behind the task selection for LoL models is not clearly articulated… it is not immediately clear why these tasks are important or how they contribute to advancing the field …
> > 4. The practical usage of the LoL models is not explicitly addressed. Are there any potential real-world applications except the dataset size prediction?
>
> Good point, we could have been clearer on how we explain the usefulness of LoL models and these tasks. See the general comment for an in-depth explanation of the usefulness. In short, there are useful applications of the tasks we consider (e.g. 20,000 times faster evaluation of finetuned models using LoL models for accuracy prediction), and we believe the LoL framework and models we introduced could be quite useful in future application work.
>
>
> > 3. Other methods used for comparison in this work are not clearly defined and lacks sufficient discussion.
>
> Could you elaborate on what you mean here? To be clear, the 5 LoL models we experiment with are proposed and implemented by us. Though, there are some similarities to existing work (e.g. we mention in Section 3.2 that our MLP+SVD is similar to the model of Salama et al. 2024, except they use a NN classifier instead of an MLP).
>
>
> > 1. The LoRA model datasets used in this work are derived from Stable Diffusion V1.4 and Qwen2. A key question is whether the LoL model will remain effective if LoRA models are obtained from different versions, such as Stable Diffusion V2.1 or Qwen V2.5.
>
> Good question. To answer it, we trained additional datasets (with Stable Diffusion 1.5, 2.1, and Llama 3.2 3B) and ran new experiments:
>
> For LLMs, we create a dataset Llama-ARC-LoRA, of 2000 finetuned Llama3.2 3 billion parameters LLMs. These networks are 2x larger and have different architecture from the Qwen2-1.5B models. Nonetheless, the results are very qualitatively similar:
>
> | Model    | Data Membership (Test Acc) | Val Loss (Test R^2) |
> |----|----|---|
> | MLP     | .522 ± .008   | .091 ± .030    |
> | MLP + O-Align    | .620 ± .007      | .562 ± .103         |
> | MLP + SVD     | .560 ± .008     | .998 ± .000       |
> | MLP + Dense     | OOM      | OOM          |
> | GL Net    | .652 ± .006   | .995 ± .000      |
>
> For Stable Diffusion, we also test OOD generalization between Stable Diffusion V1.4 and V1.5 with GLNet to show our results are not specific to Stable Diffusion V1.4.
>
> | | Trained on SD1.4 | Trained on SD1.5|
> | --- | --- | --- |
> | Test on SD1.4 |  .264 ± .006, 89.3 ± .3 | .297 ± .004, 88.1 ± .1 |
> | Test on SD1.5 |  .269 ± .004, 89.1 ± .1 |  .269 ± .005, 89.2 ± .2|
>
> We also see that our models perform well on SD2.1, though less well than for SDv1 versions:
> | Model  | Test Loss   | Test Acc  |
> | --- | --- | --- |
> | GLNet | .542 ± .013 | 74.1 ± .9 |
>
>
> > 2. Is there a clear rationale behind the choice of model architecture for the LoL models? Specifically, is using an MLP sufficient for the tasks at hand, or could other architectures be more effective?
>
> Good question. We just used an MLP for our models (besides GL-Net) for simplicity, given that we already propose 5 LoL models. We think that other architectures could indeed be better in certain ways, and are worth exploring in future work. For instance, one could try using convolutional or attentional modules, where the sequence dimension is the layer index.

---

> > ### Author Response · Authors · 2024-11-22
> >
> > Hello reviewer snsJ,
> >
> > We are checking back in to ask if you have received our response to your questions. As the response period ends on November 26, we would appreciate if you could ask any further questions now, so that we have time to answer them. Please let us know if you have any more comments, and thank you again for reviewing our paper.

---

### Official Review · Reviewer_d8i6 · 2024-11-03

**Soundness:** 3
**Presentation:** 3
**Contribution:** 4
**Rating:** 8
**Confidence:** 3

**Summary:**

This paper introduces Learning on LoRAs (LoL), an approach where low-rank adaptation weights, commonly used for efficient fine-tuning of large foundation models, serve as inputs for predictive models. The authors develop several GL(r)-equivariant and invariant neural architectures that process LoRA weights for various downstream tasks, such as predicting model accuracy, analyzing finetuning attributes, and performing data membership inference. The models leverage geometric deep learning techniques like canonicalization and invariant featurization, allowing efficient performance prediction without loss of structural information. The authors validate their approach through experimentation across diverse tasks on newly generated datasets of LoRA weights, including diffusion finetunes and language models finetunes.

**Strengths:**

1 - The paper contributes to the field by capitalizing on the structural properties of LoRA weights, enabling efficient analysis and predictions without needing full model access or extensive retraining. By focusing on parameter symmetries in LoRA weights, the method facilitates tasks like predicting model performance on downstream tasks and analyzing fine-tuning attributes directly from the LoRA weights themselves without requiring compute- and parameter-heavy inferences.

2 - The mathematical rigor in defining GL-invariant and GL-equivariant properties strengthens the reliability of the model design. In particular, Theorem 1 and Theorem 2 (Section 4)​ define the necessary properties of the proposed models, providing a clear foundation for future work in this domain. By establishing these invariance properties, the authors ensure that their architectures are theoretically grounded in symmetry-aware learning. The use of canonicalization and alignment methods for symmetry processing is well-supported by previous studies in geometric learning (see Maron et al., 2019) and further extends the concepts to the LoRA domain.

3 - The paper demonstrates extensive experiments across multiple datasets (Section 5, Tables 2 and 3)​. Notably, the authors generate three novel datasets of LoRAs (Section 5.1)​, including CelebA-LoRA, Imagenette-LoRA, and Qwen2-ARC-LoRA, providing a robust basis for evaluating their models.

**Weaknesses:**

1 - The paper could benefit from a more thorough introduction to concepts like GL-invariance, O-invariance, and O-Align before discussing them within the context of Learning on LoRAs (LoL). While these concepts are central to the proposed architecture, they may be unfamiliar to some readers, especially those not specialized in geometric deep learning. Including a dedicated background section or a more extensive explanation would make the work more accessible and improve readability.

2 - GL-net’s superior performance may be partly due to a higher number of parameters or computational requirements compared to other LoL architectures. This could imply a cost-performance trade-off, where GL-net’s success on specific tasks may not be as scalable or efficient in scenarios requiring high throughput or limited compute resources. An analysis of this cost-performance trade-off would provide helpful insight into its practical applicability.

3 - The visuals, such as Figure 3 showcasing CLIP predictions, could benefit from clearer labeling and expanded captions to help interpret performance differences among models​. While the data is comprehensive, more explanatory notes within the figures would enhance accessibility for readers.

4 - The GL(r)-equivariant models, while theoretically sound, are complex and may pose practical challenges for adoption. The multi-step processing involving canonicalization, invariant featurization, and the invariant head layer adds architectural overhead (Section 3.3.3). While this complexity is justified for theoretical exploration, a discussion on trade-offs and simplifications would make it easier for practitioners to implement the approach. Previous work in weight-space learning, such as Navon et al. (2023), has shown simpler approaches that could be integrated as baselines to contrast model complexity and efficiency​.

**Questions:**

I have no major concerns and consider the paper as a strong submission, but it would be good if the authors could elaborate on the following issues:

1 - Can the proposed methodologies (such as GL-equivariant architectures) be directly applied to other LoRA-based fine-tuning variants, like DoRA (Liu et al., 2024)? If modifications are necessary for such variants, what aspects of the approach would need adaptation, particularly in handling their unique low-rank structures?


2 - Is the performance boost primarily due to the model’s increased number of parameters or computational requirements compared to other LoL techniques (Section 5.2, Table 2)? If so, how would you describe the cost/performance trade-off for GL-net in practical applications, especially in contexts where computational efficiency is prioritized?

3 - In Table 4, GL-net shows relatively lower performance on the dataset size prediction task. Could this lower performance be attributed to overfitting? If so, would any regularization techniques be effective here, and are there any architectural adjustments that could improve generalization on this specific task?


4 - The authors briefly mention future applications of GL-equivariant networks, such as model merging and LoRA editing (Conclusion)​. Could authors please elaborate on the specific challenges or potential approaches you foresee for these applications? In particular, how might the GL-net architecture be adjusted or extended for these tasks, which may have different requirements compared to the predictive tasks presented?


References

 1- Maron, Haggai, et al. "Invariant and equivariant graph networks." arXiv preprint arXiv:1812.09902 (2018).

 2 - Navon, Aviv, et al. "Equivariant architectures for learning in deep weight spaces." International Conference on Machine Learning. PMLR, 2023.

3- Liu, Shih-Yang, et al. "Dora: Weight-decomposed low-rank adaptation." arXiv preprint arXiv:2402.09353 (2024).

---

> ### Author Response · Authors · 2024-11-18
>
> We thank the reviewer for appreciating our framework, theory, and extensive experiments.
>
> > 1 - The paper could benefit from a more thorough introduction to concepts like GL-invariance, O-invariance, and O-Align …
>
> Thank you for this comment. We will try to make our exposition clearer, and also add more background and references on relevant materials in the appendix.
>
> > 2 - GL-net’s superior performance may be partly due to a higher number of parameters or computational requirements compared to other LoL architectures…
>
> Actually, GL-net uses less parameters in general than the MLP-based methods, especially MLP + Dense. For instance, see this table (which we will include in the revised PDF).
>
> | Dataset | Model | Num params |
> | --- | --- | --- |
> | Imagenette-LoRA | MLP | 102m |
> | | MLP + O-Align | 102m |
> | | MLP + SVD | 500k |
> | | MLP + Dense | 748m |
> | | GL-Net | 40m |
> | Qwen2-ARC-LoRA | MLP  |  103m  |
> | | MLP + O-Align  | 103m |
> | | MLP + SVD  | 125k |
> | | MLP + Dense  | 352m |
> | | GL-Net | 5m |
>
> Also, as we show in Section 5.5, the preprocessing and forward pass time of GL-Net doesn’t increase much as input data size increases, but the same does not hold for some of the other baselines. However, GL-Net can be slower for smaller input networks.
>
> > 3 - The visuals … could benefit from clearer labeling and expanded captions to help interpret performance differences among models​. While the data is comprehensive, more explanatory notes within the figures would enhance accessibility for readers.
>
> Fair point. We will add additional notes, including references to relevant parts of the main text in Figure 3 and Figure 2.
>
> > 4 - The GL(r)-equivariant models, while theoretically sound, are complex … While this complexity is justified for theoretical exploration, a discussion on trade-offs and simplifications would make it easier for practitioners to implement the approach.
>
> > Previous work in weight-space learning, such as Navon et al. (2023), has shown simpler approaches that could be integrated as baselines to contrast model complexity and efficiency​.
>
> Indeed, this is why we developed 4 MLP-based LoL models in addition to GL-Net. Much like Navon et al. (2023), we proposed and empirically tested a pure MLP model, as well as an MLP with alignment. We will add more to our revised PDF of practical trade-offs, so that we elaborate more on the theoretical properties listed in Table 1.
>
> > 1 - Can the proposed methodologies (such as GL-equivariant architectures) be directly applied to other LoRA-based fine-tuning variants, like DoRA (Liu et al., 2024)?
>
> We already discussed the DoRA and KronA LoRA variants in Appendix E, with a characterization of their symmetries. In the revised PDF, we will also include additional notes on how LoL models can be applied to these variants.
>
> > 2 - Is the performance boost primarily due to the model’s increased number of parameters or computational requirements … how would you describe the cost/performance trade-off for GL-net in practical applications, especially in contexts where computational efficiency is prioritized?
>
> See number of parameters in our above answer. We are not yet sure of the full trade-offs of GL-Net, but we suspect it may be more beneficial in tasks where more expressiveness is needed, whereas the simpler MLP-based methods may be fine in other tasks.
>
> > 3 - GL-net shows relatively lower performance on the dataset size prediction task. Could this lower performance be attributed to overfitting? If so, would any regularization techniques be effective here, and are there any architectural adjustments that could improve generalization on this specific task?
>
> Yes, that is possible, and there are interesting questions here. Even though GL-net and the other MLP-based methods are expressive enough to express MLP + SVD, overfitting or some other phenomena prevent this in practice. In Section D.3 we further analyze the predictions of GL-net on this task. Perhaps different loss functions may help on this task, including regularization, or data augmentation that does not affect singular values.
>
> > 4 - The authors briefly mention future applications of GL-equivariant networks, such as model merging and LoRA editing (Conclusion)​. Could authors please elaborate…
>
> Indeed, these future applications are quite interesting. In this paper we only explored invariant tasks. To do equivariant tasks, we need to remove the invariant head of the model. Further, we may need to create different equivariant modules, such as equivariant normalization layers, which may take similar form to our equivariant nonlinearities. Model merging may be done via approaches similar to previous work on aligning model weights with metanetworks (Navon et al. 2023b), and LoRA editing may be done similarly to the INR editing tasks of Zhou et al. 2024b, where a metanetwork predicts an edit direction in a way that is trained by a supervised learning task. These require different types of losses and data setups.

---

> > ### Comment · Reviewer_d8i6 · 2024-11-21
> >
> > I thank the authors for their response. I have no further concerns and will keep my positive score.

---

### Author Response · Authors · 2024-11-18
**General Response**

Dear reviewers,

Thank you for your work in reviewing our paper. In this general comment, we address topics that apply to several reviews at once.

## Applications of LoL

The reviewers asked for further elaboration on applications of LoL models. To reiterate, we are the first to formulate the Learning on LoRAs (LoL) framework, and we created some of the first datasets of LoRAs and introduced some of the experimental tasks in our paper. We feel that LoL will be useful in applications, because:
1. **Fast evaluation.** We showed that LoL models can efficiently and accurately evaluate finetuned foundation models. For instance, GL-Net can evaluate the CLIP score of a diffusion model 53,700 times faster than standard evaluation, or the ARC-C test accuracy of an LLM 730,000 faster than standard evaluation. Standard evaluation of the LLM takes thousands of forward passes of a few billion parameter model, whereas evaluation with an LoL model takes one forward pass through a small LoL model. Fast evaluation can be used for quickly iterating on hyperparameters, and monitoring performance during training.
2. **Improving on existing works.** As mentioned in our paper’s Section 2, two works have considered special instances of learning on LoRAs: Salama et al. 2024 use nearest neighbor predictors to predict finetuning dataset size from LoRA weights, and Dravid et al. 2024 use PCA in LoRA weight space to explore finetuned diffusion models. Each of these papers have lots of motivation for their LoL tasks; we improve these applications by generalizing the framework, and developing our novel LoL models.
3. **Other metanetwork tasks.** Metanetworks (neural networks that take in other neural networks as input) have been widely studied (see pdf Section 2). Our LoL models are metanets specialized for LoRA inputs, so they can be used for any metanet application from previous work: these include generative models for generating neural network weights (Erkoc et al. 2023), learned optimization rules (Metz et al. 2022), editing neural network weights (Zhou et al. 2024b), hyperparameter optimization (Mehta et al. 2024), or domain adaptation (Navon et al. 2023a). While we could not fit in all of these applications, our LoL models allow future work to apply metanets to this new domain of LoRA weights.
4. **Interpretability via LoL models.** Another promising avenue is interpreting neural network weights via LoL models. For instance, in Imagenette classification, an LoL model gets 90% accuracy when training only on LoRAs of attention query matrices, but it gets only 66% accuracy when training only on LoRAs of attention key matrices. This implies that finetuning query matrices is more important for this task than key matrices. Also, we explore interpreting LoL model predictions in dataset size prediction in Section 5.2.2 and Appendix D.3.

## Other contributions
We are the first to characterize the set of GL-Equivariant linear maps and prove their completeness (Proposition 1). Equivariant linear maps are the essential building block of equivariant architectures, as outlined in the Geometric Deep Learning Blueprint (Bronstein et al. 2021), and used for different domains by Zaheer et al. 2017, Finzi et al. 2021, Navon et al. 2023, and many others. Our contributions of characterizing GL-equivariant linear maps (and other equivariant/invariant modules) can be useful to future work, both for Learning on LoRAs and for other domains with GL-symmetries: such as eigenvector processing for non-symmetric matrices (Lim et al. 2022), or metanetworks for linear networks (Kunin et al. 2019)
In Section 5.4 we show zero-shot OOD generalization of our LoL models to different ranks. This is in part allowed by the invariance properties of our models.
As elaborated on in Section 5.5, while existing metanetworks may deal with input networks of a few thousand or million parameters, we process networks of billions of parameters, by only processing their LoRA weights. Moreover, we show in Section 5.5 that our LoL models can scale to 175B parameters (GPT-3 size). Thus, our framework allows us to greatly expand the applicability of metanetworks.

**References:**
* (Salama et al. 2024) Dataset Size Recovery from LoRA Weights
* (Dravid et al. 2024) Interpreting the weight space of customized diffusion models.
* (Erkoc et al. 2023) HyperDiffusion: Generating Implicit Neural Fields with Weight-Space Diffusion
* (Metz et al. 2022) Velo: Training versatile learned optimizers by scaling up.
* (Mehta et al. 2024) Improving Hyperparameter Optimization with Checkpointed Model Weights
* (Bronstein et al. 2021) Geometric Deep Learning: Grids, Groups, Graphs, Geodesics, and Gauges
* (Zaheer et al. 2017) Deep Sets
* (Finzi et al. 2021) A Practical Method for Constructing Equivariant Multilayer Perceptrons for Arbitrary Matrix Groups
* (Kunin et al. 2019). Loss Landscapes of Regularized Linear Autoencoders. ICML 2019.

---

### Meta-Review · Area_Chair_ySfF · 2024-12-21

**Metareview:**

**Summary**


The paper introduces Learning on LoRAs (LoL), which performs regression/classification tasks on the low-rank adaptation parameters of a base model. To enable parameteric learning on the LoRA weights, the authors consider the inherent symmetries of these weights and propose various GL(r)-equivariant/invariant models to perform learning on these weights. Such parameteric model could be used for various downstream tasks, such as predicting model accuracy, analyzing finetuning attributes, and performing data membership inference. The authors validate their approach through experimentation across diverse tasks on newly generated datasets of LoRA weights, including LoRA adaptation paramters of diffusion and language models.


**Strengths**


* The paper proposes a novel problem, that could become of interest to the community.
* The authors generate three novel datasets of LoRAs​, including CelebA-LoRA, Imagenette-LoRA, and Qwen2-ARC-LoRA, providing a robust basis for evaluating their models. If publicly released, such dataset would be of interest to the community.
* The careful treatment of the required invariances for designing the LoL models and the proposed different options are interesting.


**Weaknesses**

* The paper does not provide clear details about the models, including the number of parameters and the training costs associated with various model structures and methods.
* While the proposed problem is interesting from a research perspective, its practical implications are not well motivated. In particular, LoL models aim to predict hyperparameters, CLIP scores, and training images of fine-tuned diffusion models using their LoRA weights. However, it is not immediately clear why these tasks are important or how they contribute to advancing the field.
* It is not clear, whether the propose method can generalize across different base models (e.g., same architecture with different  parameters).


**Conclusion**

he paper received polarized reviews. Reviewer EBmP offered a low-quality and dismissive critique, rating it 3 with a confidence of 3. Reviewer d8i6 was more positive, giving it an 8, though also with low confidence (confidence 3). Reviewer snsJ provided a more robust and confident review (rating 5, confidence 4), highlighting significant issues. The authors countered with a substantial rebuttal that addressed some concerns regarding generalization across base models. After thoroughly reviewing the paper, the other reviews, and the authors' rebuttal, my assessment aligns with Reviewer snsJ, finding the paper just below the acceptance threshold. The topic is compelling, but the practical implications are not convincingly presented, a concern also noted by Reviewer EBmP, despite the overall poor quality of their review. The inclusion of more concrete practical applications, such as using the method to detect if an adapted model is compromised (e.g., the training data contained backdoors), would significantly strengthen the paper. Given these considerations, I vote to reject this paper.

**Additional Comments On Reviewer Discussion:**

This paper received highly polarized reviews. Despite my repeated attempts to engage the reviewers in discussions during the review and discussion periods to reach a consensus on the paper's merits and shortcomings, there was no participation in any discussions. This made the evaluation of this paper very challenging.

After thoroughly reading the paper and considering the reviewers' feedback, I find the paper to lack proper motivation for the proposed method. Hence, I vote for rejecting the paper.

---

### Decision · Program_Chairs · 2025-01-22

Reject